# The role of Cdc42 and Gic1 in the regulation of septin filament formation and dissociation

Yashar Sadian[1†], Christos Gatsogiannis[1†], Csilla Patasi[2], Oliver Hofnagel[1], Roger S Goody[1], Marian Farkašovský[2], Stefan Raunser[1]*

[1]Department of Physical Biochemistry, Max Planck Institute of Molecular Physiology, Dortmund, Germany; [2]Department of Molecular Microbiology, Institute of Molecular Biology SAS, Bratislava, Slovak Republic

**Abstract** Septins are guanine nucleotide-binding proteins that polymerize into filamentous and higher-order structures. Cdc42 and its effector Gic1 are involved in septin recruitment, ring formation and dissociation. The regulatory mechanisms behind these processes are not well understood. Here, we have used electron microscopy and cryo electron tomography to elucidate the structural basis of the Gic1-septin and Gic1-Cdc42-septin interaction. We show that Gic1 acts as a scaffolding protein for septin filaments forming long and flexible filament cables. Cdc42 in its GTP-form binds to Gic1, which ultimately leads to the dissociation of Gic1 from the filament cables. Surprisingly, Cdc42-GDP is not inactive, but in the absence of Gic1 directly interacts with septin filaments resulting in their disassembly. We suggest that this unanticipated dual function of Cdc42 is crucial for the cell cycle. Based on our results we propose a novel regulatory mechanism for septin filament formation and dissociation.

*For correspondence: raunser@mpi-dortmund.mpg.de

†These authors contributed equally to this work

Competing interests: The authors declare that no competing interests exist.

## Introduction

Septins are ubiquitous guanine nucleotide-binding proteins that have been implicated in many cellular processes such as cytokinesis, spindle positioning, morphogenesis, and exocytosis, and their mutation or overexpression is associated with neoplasia, neurodegenerative diseases and male infertility (*Hall and Russell, 2004*).

In yeast, four essential septins (Cdc3, Cdc10, Cdc11 and Cdc12) are found at the bud neck (*Haarer and Pringle, 1987*; *Ford and Pringle, 1991*; *Kim et al., 1991*), where they form an ordered ring composed of membrane-adjacent filaments (*Hartwell, 1971*; *Byers and Goetsch, 1976*). In total seven different septins were identified in *S. cerevisiae*, where they form filaments of variable size and combinations. Whereas the human genome encodes thirteen septins, *C. elegans* has only two and plants are devoid of septin genes (*Hall and Russell, 2004*; *Ihara et al., 2005*; *Kinoshita, 2006*). Despite the genetic variability, all septins share defined structural features. A recent crystallographic study on the human SEPT2-SEPT6-SEPT7 complex has shed light on the structural organization of human septins at the atomic level, which differs profoundly from that of other cytoskeletal structures (*Sirajuddin et al., 2007*, *2009*). Septins interact via their central guanine nucleotide-binding domains (G-domains) and/or the N- and C-terminal extensions forming oligomers and non-helical filaments.

The basic structural unit of the yeast septin complex is an octamer, composed of four subunits, namely Cdc10, Cdc3, Cdc12 and Cdc11, arranged into two tetramers with two-fold rotational symmetry (*Bertin et al., 2008*).

Cdc42 has been identified as a central regulator of septin ring assembly and disassembly during different stages of the cell cycle (*Gladfelter et al., 2002*; *Kozminski et al., 2003*). Mutations that affect

**eLife digest** Septins are proteins that provide structural support for cells as they divide. Yeast cells are known to have seven types of septins, which have been widely studied, and 13 different septins have been identified in human cells, although they all seem similar to those found in yeast. Mutations in the genes that carry the genetic code for septins lead to a range of debilitating conditions in humans, including neurodegenerative diseases and male infertility.

An enzyme called Cdc42 is thought to have a key role in the formation of ring-like structures by septins before a cell divides, and in the subsequent dismantling of these rings after the cell has divided. A pair of proteins, called Gic1 and Gic2, is known to be critical for the formation of the septin rings, but the details of the interactions between these two proteins, Cdc42 and the septins are sketchy.

Now Sadian et al. have used two imaging approaches—electron microscopy and cryo-electron tomography—to scrutinise the role of Gic1 in greater detail in yeast cells. Gic1 interacts with specific subunits within adjacent septins, and these interactions have the effect of crosslinking the septins and stabilizing them in long filaments. However, high concentrations of the enzyme Cdc42 block the interaction between the Gic1 proteins and the subunits, causing the filaments to be dismantled. A future challenge will be to elucidate the interaction of these proteins in molecular detail using other techniques, in particular X-ray crystallography.

the GTPase activity of Cdc42 impair the initial assembly of septin rings, while after bud emergence, septin rings are maintained independently of Ccd42 (*Gladfelter et al., 2002*). It was also reported that the activity of Cdc42's guanine nucleotide exchange factor (GEF) and GTPase activating protein (GAP) are required for proper septin ring formation and localization, implying that one or more cycle(s) of nucleotide binding and hydrolysis are required for Cdc42 at the beginning of budding (*Gladfelter et al., 2002*; *Caviston et al., 2003*).

Among the essential effectors of Cdc42 in yeast are the two structurally homologous proteins Gic1 and Gic2, which are functional homologues of the human Borg protein (*Joberty et al., 2001*; *Sheffield et al., 2003*). It has been shown that Gic1 and Gic2 play an essential and overlapping role in cytoskeletal polarization (*Brown et al., 1997*; *Hall and Russell, 2004*) and septin recruitment (*Iwase et al., 2006*). However, the complex interplay between Cdc42, Gic1 and septins at the molecular level and its role during the cell cycle is not yet understood.

In this study, we have used electron microscopy and cryo electron tomography (cryo-ET) to describe the structural basis for the direct interaction of Gic1 and Cdc42 with septin filaments. Gic1 interacts with Cdc10 subunits of adjacent septin filaments and cross-links them. Because of this scaffolding, septin filaments are stabilized and form long railroad-like ordered filament cables. Cdc42-GTP directly binds to Gic1 and at higher concentrations inhibits the Gic1 interaction with Cdc10, resulting in the dissociation of the Gic1-septin complex. In its GDP-state, however, in absence of Gic1 Cdc42 interacts directly with Cdc10 and thereby mechanically disassembles septin filaments. Gic1 and Cdc42-GDP therefore compete for the same septin subunit. Finally, based on our results we propose a novel regulatory mechanism for septin ring formation and dissociation involving Cdc42 and Gic1.

## Results and discussion

EGFP-labeled septins without Gic1 (Cdc3-EGFP, Cdc10, Cdc11 and Cdc12) form relatively short and straight filaments (*Figure 1A*). Interestingly, when Gic1 is added during septin polymerization, long filaments that cluster together in large bundles are formed (*Figure 1B*). Studying the same but not EGFP-labeled samples using electron microscopy (EM), we found that in contrast to blank septin polymers that form long, often pairwise arranged filaments (*Figure 1C*), Gic1-septin complexes display a regular railroad-like structure with many cross-linked filaments bundled together (*Figure 1D*). Gic1 forms cross-bridges between at least two filaments, keeping them at a distance of about 20 nm (*Figure 1E*). At each cross-bridge, Gic1 binds to at least two adjacent septin subunits on each filament (*Figure 1F*), leaving a gap of six free subunits between individual Gic1 molecules (*Figure 1G*). The structure of Gic1 is not well defined (*Figure 1F*) indicating that Gic1 is flexible and oriented differently at each cross-bridge.

To determine which septin subunit interacts with Gic1, we labeled septin-Gic1 complexes with antibodies against Cdc11 and observed that it sits exactly in the middle of a septin filament between two

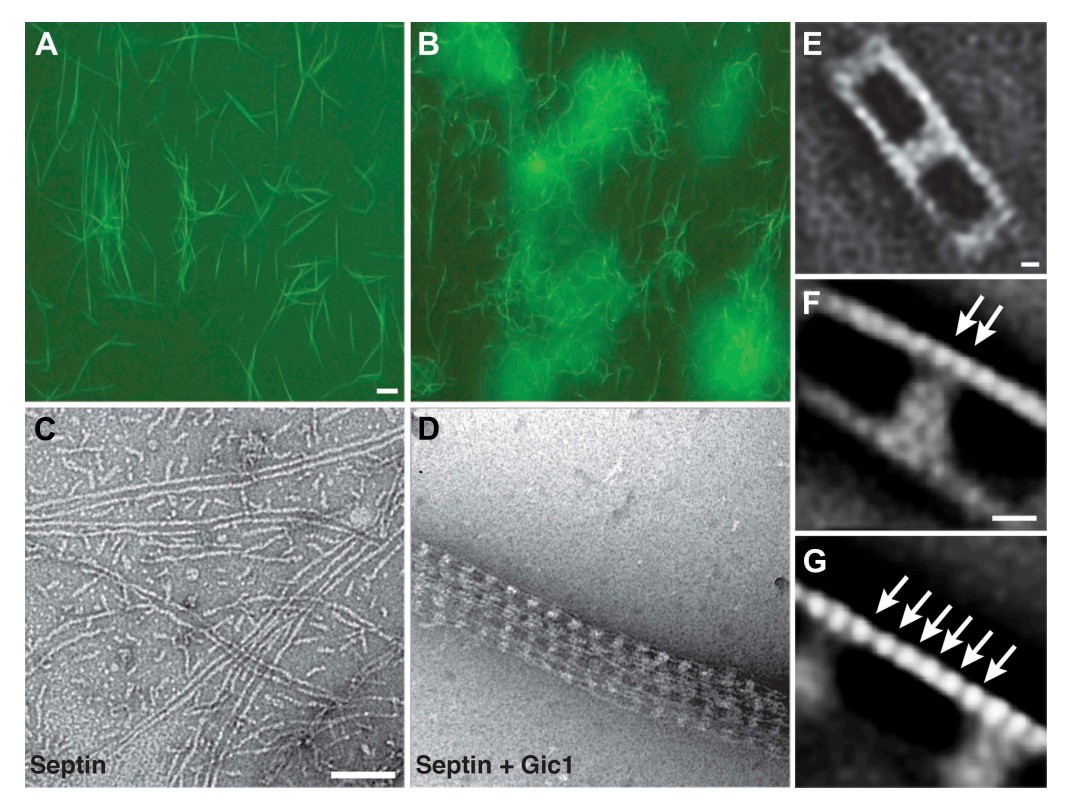

**Figure 1**. Gic1 scaffolds septin filaments resulting in long and flexible filament cables. (**A** and **B**) Yeast septin octamers containing Cdc3-EGFP polymerized by dialysis alone (**A**) or together with Gic1 (**B**) and imaged using fluorescence microscopy. Scale bar, 0.5 μm. (**C** and **D**) Representative EM image of negatively stained septin filaments (**C**) and septin-Gic1 complexes (**D**) without EGFP. Scale bar, 100 nm. (**E–G**) Representative class averages with focus on the overall structure of the septin-Gic1 complex (**E**), the Gic1 cross-bridges (**F**) and the septin filaments (**G**); arrows indicate single septin proteins. Scale bars, 10 nm.

Gic1 cross-bridges (*Figure 2A,B*). Based on the known sequential order of septin filaments, this suggests that Gic1 binds to Cdc10 (*Figure 2C*).

However, previous yeast two-hybrid, in vitro interaction and coimmunoprecipitation data indicated that Gic1 directly interacts with Cdc12 (*Iwase et al., 2006*). Therefore, we performed additional experiments to further corroborate our findings and prepared septin-Gic1 complexes devoid of either Cdc10 or Cdc11 (*Figure 2D–I*).

As expected from the results of antibody labeling, septin-Cdc10Δ, which formed short filaments, did not bind to Gic1 (*Figure 2D–E*). Septin-Cdc11Δ, however, which in accordance with previous studies (*Bertin et al., 2010*) does not polymerize (*Figure 2F*) produced a similar track-like structure as the wildtype when Gic1 was added (*Figure 2G*). Thus, addition of Gic1 resulted not only in the binding of the protein to the septin complex but also induced septin polymerization. We then performed single particle analysis (SPA) of the septin filaments and observed that only four subunits were located between Gic1 cross-bridges of Cdc11Δ filaments, ruling out an interaction with Cdc12 (*Figure 2H–I*).

In addition, yeast two-hybrid studies supported our in vitro data, showing that Gic1 only interacts with Cdc10 (*Figure 2J*). Notably, Cdc10 is mainly responsible for the specific interaction of septin filaments with PIP2, localizing them to membranes and promoting filament polymerization and organization (*Bertin et al., 2010*). For Gic2, which is a homologue of Gic1, a direct interaction through its polybasic region with PIP2 was also reported (*Orlando et al., 2008*). Together with our findings, this suggests that the organization and polymerization of septin filaments is controlled by the interaction of Gics with Cdc10 and the interaction of both proteins with PIP2.

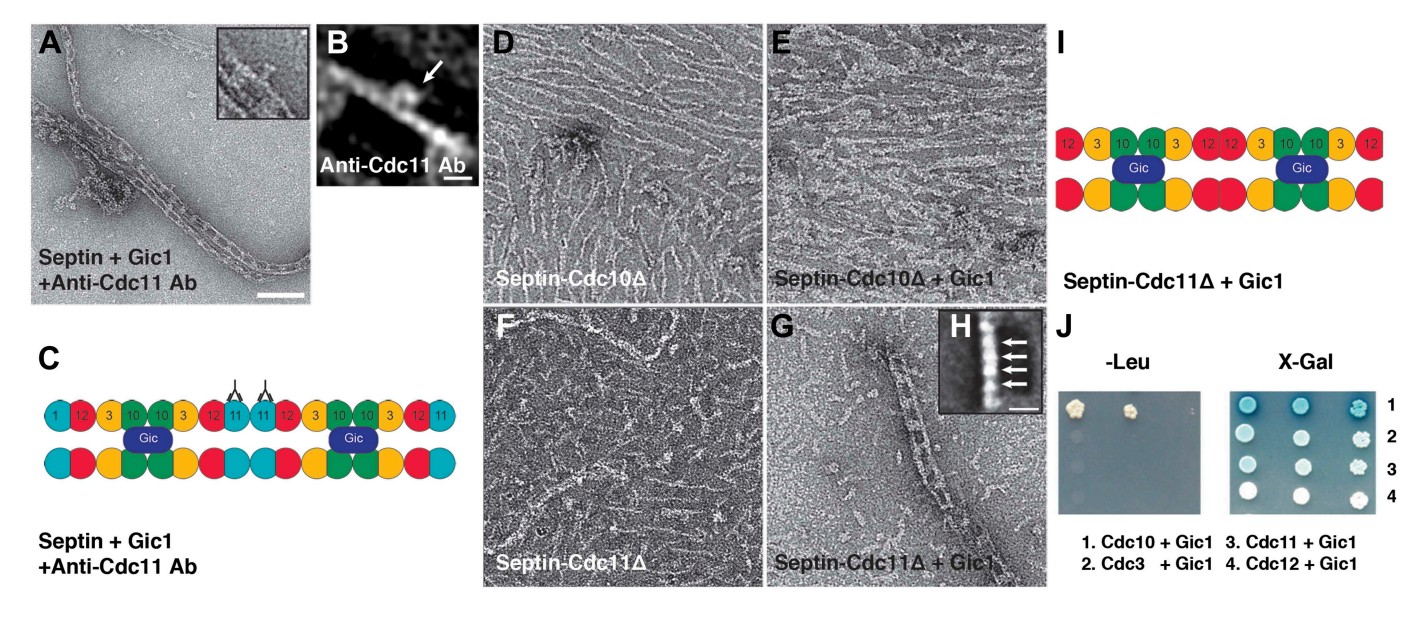

**Figure 2**. Gic1 binds specifically to septin Cdc10. (**A** and **B**) Representative EM image (**A**) and class averages (**B**) of septin-Gic1 complexes labeled with antibody against Cdc11. Arrow indicates the antibodies. The class average in (**B**) contains 15 single particles. Scale bars, 100 nm and 10 nm in (**A**) and (**B**), respectively. (**C**) Model of the septin-Gic1 complex based on the known sequential order of septin filaments (*Bertin et al., 2008*). The G- and the N/C-interfaces are indicated by straight and circular interfaces between circles, respectively. Antibodies are indicated as Y shapes. (**D** and **E**) Representative EM images of septin-Cdc10Δ filaments without (**D**) and with Gic1 (**E**). (**F** and **G**) Representative EM images of septin-Cdc11Δ filaments (**F**) and septin-Cdc11Δ-Gic1 complexes (**G**). (**H**) Representative class average of the septin-Cdc11Δ-Gic1 complex with focus on the septin filament. Arrows indicate single septin proteins. Scale bar, 10 nm. (**I**) Model of the septin-Cdc11Δ-Gic1 complex. (**J**) Yeast two-hybrid assay of different septin proteins, namely Cdc3, Cdc10, Cdc11, and Cdc12 with Gic1.

Bertin et al. reported that septin filaments are cross-linked by overlapping C-terminal extensions of Cdc3 and Cdc12 (*Bertin et al., 2008*). Depending on the stain thickness we also observed these thin cross-links between bare septin filaments. However, in septin-Gic1 complexes, the large Gic1 cross-bridges are very prominent, causing stain to accumulate between the Gic1 cross-bridges. This makes it impossible to visualize the thin coiled-coils between Cdc3 and Cdc12, although they are probably still there.

To study the native three-dimensional structure of the filaments, we vitrified septin-Gic1 complexes and determined their structure using cryo electron tomography (cryo-ET) (*Figure 3*; *Videos 1–4*). We observed that Gic1, instead of cross-bridging only two filaments, simultaneously interacted with up to six septin polymers, forming long filament cables (*Figure 3E–G*). The organization of these cables is such that Gic1 forms a central scaffold to which septin filaments attach. Remarkably, individual septin filaments do not start at the same position and run along in parallel rigid lines. The septin-Gic1 cables are rather composed of many filaments of variable lengths that start at random positions, sometimes bypassing a Gic1 cross-bridge or changing place with another filament (*Figure 3H–I*). Filaments are often only attached to adjacent filaments for several nanometers. This gives the septin-Gic1 cables a certain flexibility that on the one hand allows them to bend and adjust to membrane curvature (*Figure 3C*). On the other hand it enables the interaction between cables, resulting in mesh-like structures (*Figure 3D*). Because of the limited resolution of the reconstructions, the additional thin connections described by Bertin et al. (*Bertin et al., 2008*) are not visible.

Both in three-dimensional reconstructions and two-dimensional class averages, the density corresponding to Gic1 is poorly defined, suggesting a flexible structure of Gic1. Gic1, which based on its sequence has a molecular weight of 23 kDa (Gic1(104-314)) (*Figure 4A*), elutes at about 49 kDa from a gel filtration column corresponding roughly to a dimer (*Figure 4B*). In a typical Gic1 cross-bridge up to 12 Cdc10 subunits are involved (two per filament) (*Figure 3E*). If we assume that each of them binds independently to a Gic1 dimer, we would expect that 12 Gic1 dimers assemble into a large cross-bridge of 600 kDa. Although we cannot exclude that the density corresponding to Gic1 appears

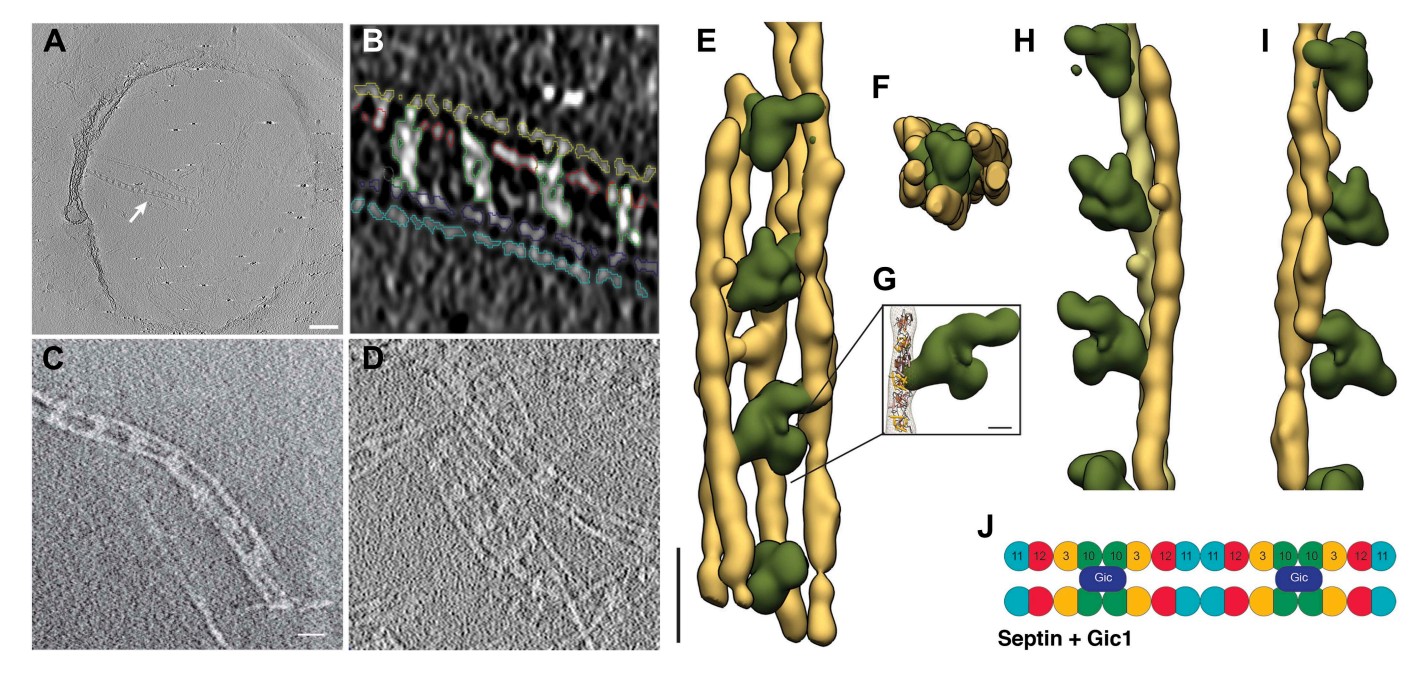

**Figure 3**. Cryo-ET of the septin-Gic1 complex. (**A**) Central slice of a cryo electron tomogram (for full tomogram see **Videos 1–3**). Arrow indicates a septin-Gic1 cable. Scale bar, 200 nm. (**B**) Segmentation of the tomograms. (**C** and **D**) Extracts from tomograms that show the flexibility and bending of the septin-Gic1 complex (**C**) as well as its ability to form mesh-like structures (**D**). Scale bar, 20 nm. (**E–G**) Side (**E**) and top view (**F**) of a septin-Gic1 complex. The septin filaments and Gic1 cross-bridges are depicted in gold and green, respectively. Scale bar, 20 nm. (**G**) The crystal structure of the human SEPT2/6/7 complex (PDB 2QAG) was manually fit into the EM structure. Scale bar, 5 nm. (**H** and **I**) Side views of the septin-Gic1 3D models. To allow a better observation of the septin filament interaction with Gic1, part of the septin filaments have been omitted. Scale bar, 20 nm. (**J**) Model of the septin-Gic1 complex based on the known sequential order of septin filaments (**Bertin et al., 2008**). The G- and the N/C-interfaces are indicated by straight and circular interfaces between circles, respectively.

more spread out due to missing wedge artifacts in our tomograms, the average large volume of the Gic1 density in the electron tomograms (**Figure 3E–I**) indicates that it must contain multiple Gic1 molecules. In addition, the heterogeneity of the Gic1 cross-bridges suggests that the number of Gic1 molecules varies between cross-bridges.

It was previously reported that Gic1 and its homologue Gic2 are effectors of Cdc42, which bind via their CRIB motif to the GTP-bound form of the Rho GTPase in cells (**Brown et al., 1997**; **Chen et al., 1997**). We made similar observations when studying the interaction of purified Gic1 and Cdc42 in vitro. Gic1 formed a stable complex with Cdc42-GppNHp (non-hydrolysable GTP analogue), but did not bind to Cdc42 in the GDP-state (**Figure 5**). On producing septin-Gic1 complexes in the presence of Cdc42-GppNHp, we found the same railroad-pattern as for the septin-Gic1 complexes, however, in this case with much bulkier cross-bridges (**Figure 6A**).

SPA and cryo-ET of the Gic1-Cdc42-septin complex revealed that the cross-bridges were almost one and a half times larger in size, spanning not only two but four septin subunits, leaving only four septins unoccupied (**Figure 6B–I**, **Video 5**). This suggests a direct interaction of Cdc42-GppNHp with Gic1 on septin filaments (**Figure 6J**).

Unexpectedly, an increase of the Cdc42-GppNHp concentration resulted in dissociation of the septin-Gic1-Cdc42 complex (**Figures 6D and 7**). Smaller in size, but still filamentous, the septin filament structure deteriorated somewhat as a result of this dissociation. Gic1 and Cdc42, however, formed protein aggregates (**Figure 7C,F**). A possible explanation for this observation is that full decoration of Gic1 with Cdc42 destabilizes the Gic1 interaction with septins, resulting in dissociation of the septin-Gic1-Cdc42 complex. In contrast, Cdc42-GDP did not bind to septin-Gic1 complexes even at 10 times higher concentrations (**Figure 8**). Similar behavior was reported for the functionally related human protein Borg (**Joberty et al., 1999**; **Sirajuddin et al., 2007**). Borg binds with its BD3

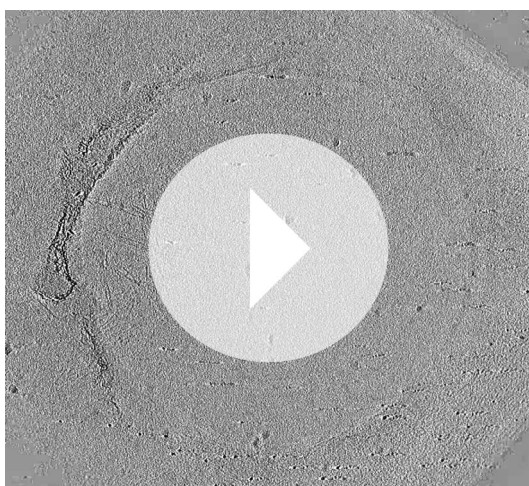

**Video 1**. Video through a cryo electron tomogram of the septin-Gic1 complex with bundles running perpendicular to the beam.

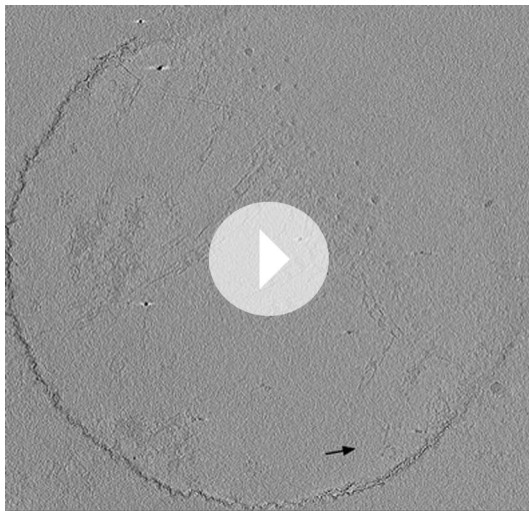

**Video 2**. Video through a cryo electron tomogram of the septin-Gic1 complex with bundles running parallel to the beam (indicated by an arrow).

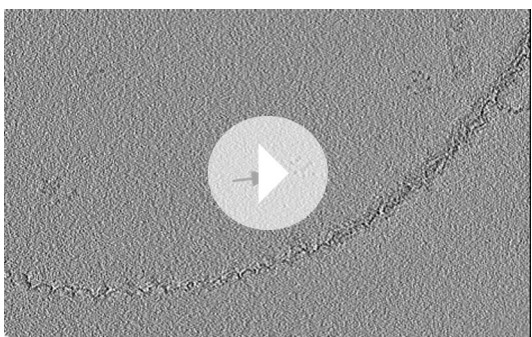

**Video 3**. Close-up on a septin-Gic1 complex running parallel to the beam (full tomogram see **Video 2**).

domain to SEPT7/SEPT6 dimers and induces the formation of long and thick septin filament fibers (**Joberty et al., 2001**; **Sheffield et al., 2003**). In accordance with our observations with Cdc42-GTP, Gic1 and septin, the human Cdc42-GTP negatively regulates this process and after binding to Borg inhibits the Borg–septin association.

To determine whether Cdc42 interacts directly with septins in the absence of Gic1, we produced septin filaments in the presence of Cdc42-GppHNp and Cdc42-GDP, respectively. Although Cdc42 in its GTP-state bound to septin filaments, no regular pattern as in the case of Gic1 was observed, indicating non-specific interaction (**Figure 9A**). Surprisingly, Cdc42-GDP bound more specifically to septin filaments, and formed, in contrast to Cdc42-GppNHp, railroad track-like structures (**Figure 9B**). Even more surprising was that Cdc42-GDP completely dissociated septin filaments at higher concentrations in less than an hour (**Figures 9C and 10**). We then analyzed the oligomers by size-exclusion chromatography, which showed that Cdc42-GDP binds to septin complexes (**Figure 9D**). SPA of the dissociated septins clearly identified either octamers with an extra density at their ends (90%) or hexamers without additional density (10%) (**Figure 9E–F**). Cdc3 sits at the ends of the hexamers, and at the second outer position of the octamers (**Figure 9G–H**). Considering the sequential order of septin oligomers, this indicates that Cdc42-GDP directly binds to Cdc10 (**Figure 9I**), which is also the binding partner of Gic1 (**Figure 2**), thereby causing dissociation of the filaments. Consequently, the resulting septin octamers differ from the ones observed under high-salt conditions (**Figure 11**). Instead of Cdc11, which is now in the center, Cdc10 forms the cap of the octamers at both ends. In addition, binding of Cdc42-GDP to Cdc10 probably influences its interaction with Cdc3 at the G interface resulting in the formation of Cdc42-GDP-Cdc10 dimers and hexameric (Cdc3-Cdc12-Cdc11) complexes. A direct interaction of a protein with Cdc42-GDP is unusual, since in most reported cases Cdc42, like other small G proteins, binds to its effectors in its active, that is GTP-bound state. However, also other proteins such as Msb3 and Msb4, which are GAP proteins, that are, like Gic1, involved in cell polarization, were shown to directly interact with Cdc42-GDP (**Tcheperegine et al., 2005**).

To identify the Cdc42 binding site on Cdc10, we calculated homology models of Cdc3, Cdc10, Cdc11 and Cdc12 using the SEPT2 structure (PDB 2QA5) as a reference and mapped regions of conservation between the four structures on

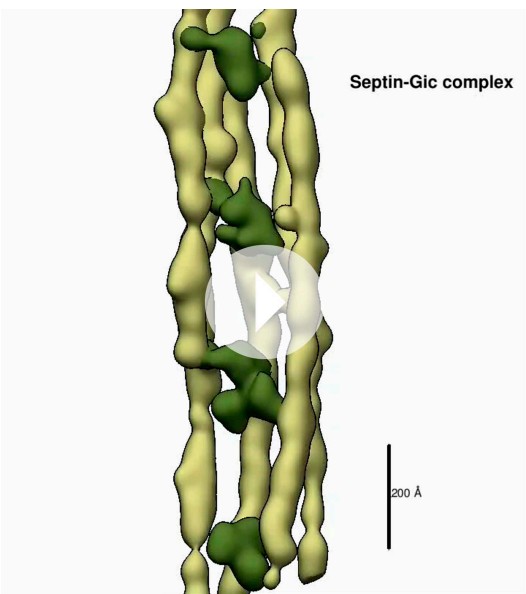

**Video 4**. Video of the 3D reconstruction of the septin-Gic1 complex derived from tomograms of vitrified samples.

their surfaces. Examining the Cdc10-Cdc10 interface, which is an N/C interface, we found that the N-terminal region is the least conserved, and would therefore present an ideal site for specific Cdc42-GDP binding (*Figure 12A–B*). We deleted the N-terminal 29 residues of Cdc10 and produced complexes whose Cdc10 was replaced by Cdc10(30-322). It was previously shown that deletion of the N-terminal residues of Cdc10 results in non-polymerizing septin complexes forming tetramers and inside-out octamers (*Bertin et al., 2010*). We analyzed the particles by EM and SPA, which showed that most of the complexes were indeed tetramers or octamers, and that the polymerization of the complexes was clearly impaired, as we did not observe filaments at low-salt conditions (*Figure 13A–C*). We then incubated the oligomers with high concentrations of Cdc42-GDP and looked for additional densities corresponding to Cdc42-GDP by EM and SPA. However, no additional protein was bound to the tetramers or octamers (*Figure 13D–F*). Gel filtration experiments corroborated this result and showed that indeed Cdc42-GDP does not bind to Cdc10(30-322) (*Figure 13G*). This indicates that the N-terminal region of Cdc10 is essential for the binding of Cdc42-GDP to septin filaments. Interestingly, the same region on Cdc10 was shown to be important for the interaction of septin filaments with PIP2 (*Bertin et al., 2010*).

From our previous experiments, we know that in septin-Gic1 complexes Cdc42-GDP binds neither to septins nor to Gic1 (*Figure 8*). Since both Gic1 and Cdc42-GDP bind to Cdc10 and very probably compete for the same binding site, Gic1 must have a higher affinity for Cdc10 than Cdc42-GDP. In order to prove this, we added Gic1 to septin-Cdc42-GDP complexes (*Figure 14A–B*). As expected, we found

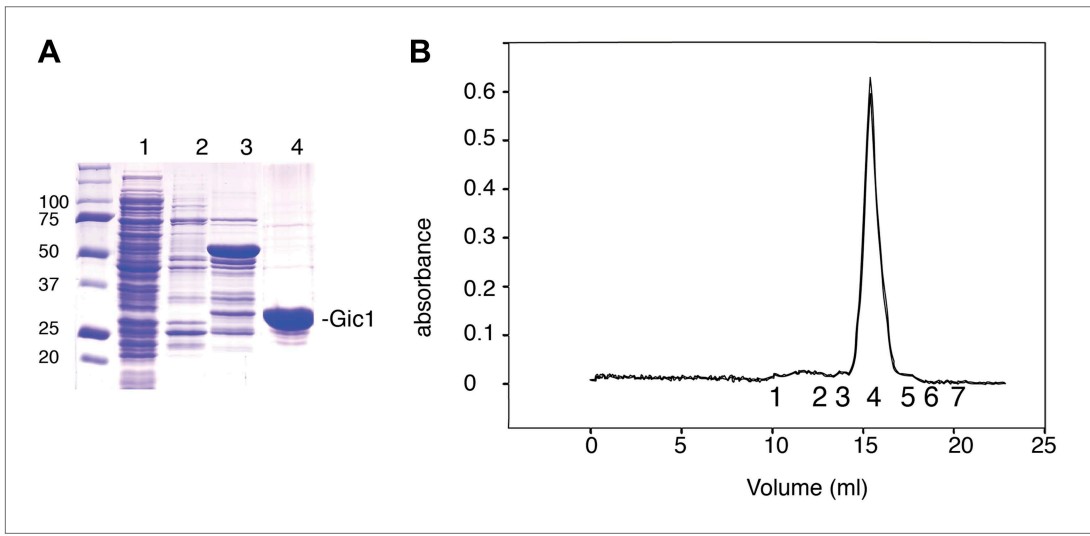

**Figure 4**. Purification of Gic1. (**A**) SDS-PAGE of the Gic1(104-314) purification. (1) Flow through Ni-NTA, (2) wash Ni-NTA, (3) elution Ni-NTA, (4) elution from gel filtration after cleavage with prescission protease. (**B**) Gel filtration chromatography of purified Gic1(104-314). The calculated molecular weight of Gic1(104-314) is 23.38 kDa. The protein elutes from the gel filtration chromatography at a volume that corresponds to a molecular weight of 49 kDa, suggesting that it forms dimers. Protein standards for gel filtration: 1, ferritin (440 kDa); 2, aldolase (158 kDa); 3, conalbumin (75 kDa); 4, ovalbumin (43 kDa); 5, carbonic anhydrase (29 kDa); 6, RNase A (13.7 kDa); 7, aprotinin (6.5 kDa).

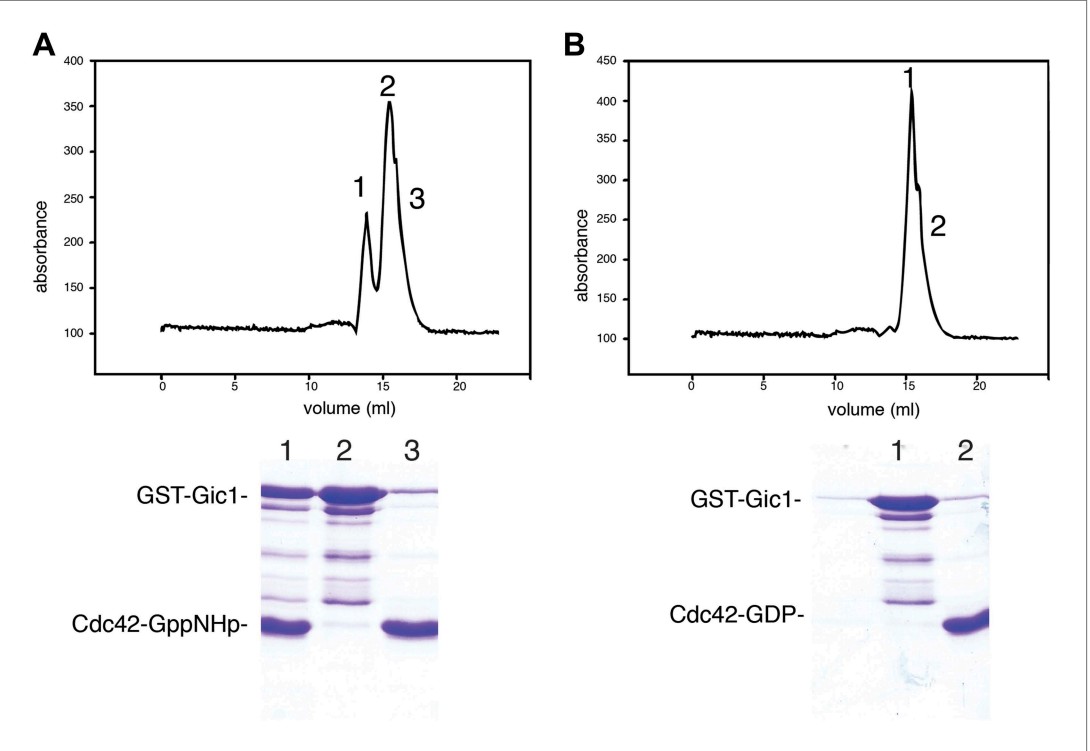

**Figure 5**. Cdc42-GppNHp binds specifically to Gic1. (**A** and **B**) Gel filtration chromatography and SDS-PAGE of Gic1 with Cdc42-GppNHp (**A**) and Cdc42-GDP (**B**), respectively.

long railroad track-like ordered filaments indicating that Gic1 replaced Cdc42-GDP to interact with Cdc10, resulting in septin polymerization.

We also observed septin-Gic1 filament complexes when we mixed Gic1 with octamers under conditions that normally inhibit polymerization, such as high-salt buffer (*Figure 14C–D*). This was surprising, since Gic1 binds and crosslinks Cdc10 proteins while high salt weakens the interaction at the N/C-interface between two Cdc11 proteins. Thus, the stabilizing effect of Gic1 must be stronger than the weakening effect of high salt. We confirmed this observation by performing the inverse experiment, that is septin-Gic1 complexes were dialyzed against a high-salt buffer. As expected, septin-Gic1 complexes were stable and did not dissociate (*Figure 14E–F*). Even septin complexes completely lacking Cdc11, which do not polymerize at low-salt conditions, form filaments when Gic1 is added (*Figure 2G*). Taken together, we conclude that Gic1 not only scaffolds septin filaments, but also increases their stability.

We then asked whether Gic1 would also stabilize septin filaments whose assembly is weakened at the Cdc10-Cdc10 instead of the Cdc11-Cdc11 interface. As shown above, complexes whose Cdc10 is replaced by Cdc10(30-322) do not polymerize at all and form tetramers or inside-out octamers (*Figure 13A–C*). Interestingly, Gic1 did not induce filament formation of such impaired septins (*Figure 13H*), but still bound to Cdc10(30-322) as indicated by an additional density attached to the septin oligomers and by size-exclusion chromatography (*Figure 13I–L*).

Our data (see *Figure 15* for overview) provide important insights into the interaction of Gic1, Cdc42 and septins. Although our analysis focuses on only three proteins that are important for septin recruitment, ring formation and disassembly, which are complex processes involving many proteins (reviewed in *Park and Bi, 2007*), we think that the new information provided by our study is coherent enough to suggest the following mechanism for the interplay between Gic1, Cdc42 and septins during budding or the cell cycle.

Because Cdc42-GDP binds to Cdc10 in septin complexes and prevents their polymerization, we propose that Cdc42-GDP rather than Gic1 recruits septin octamers (with Cdc10 at the caps) to the bud site (*Figure 16A*). At the bud site Cdc42 interacts with its GEF Cdc24, which catalyzes the exchange of its nucleotide (*Figure 16B*). As a result, Cdc42 dissociates from the septin complexes,

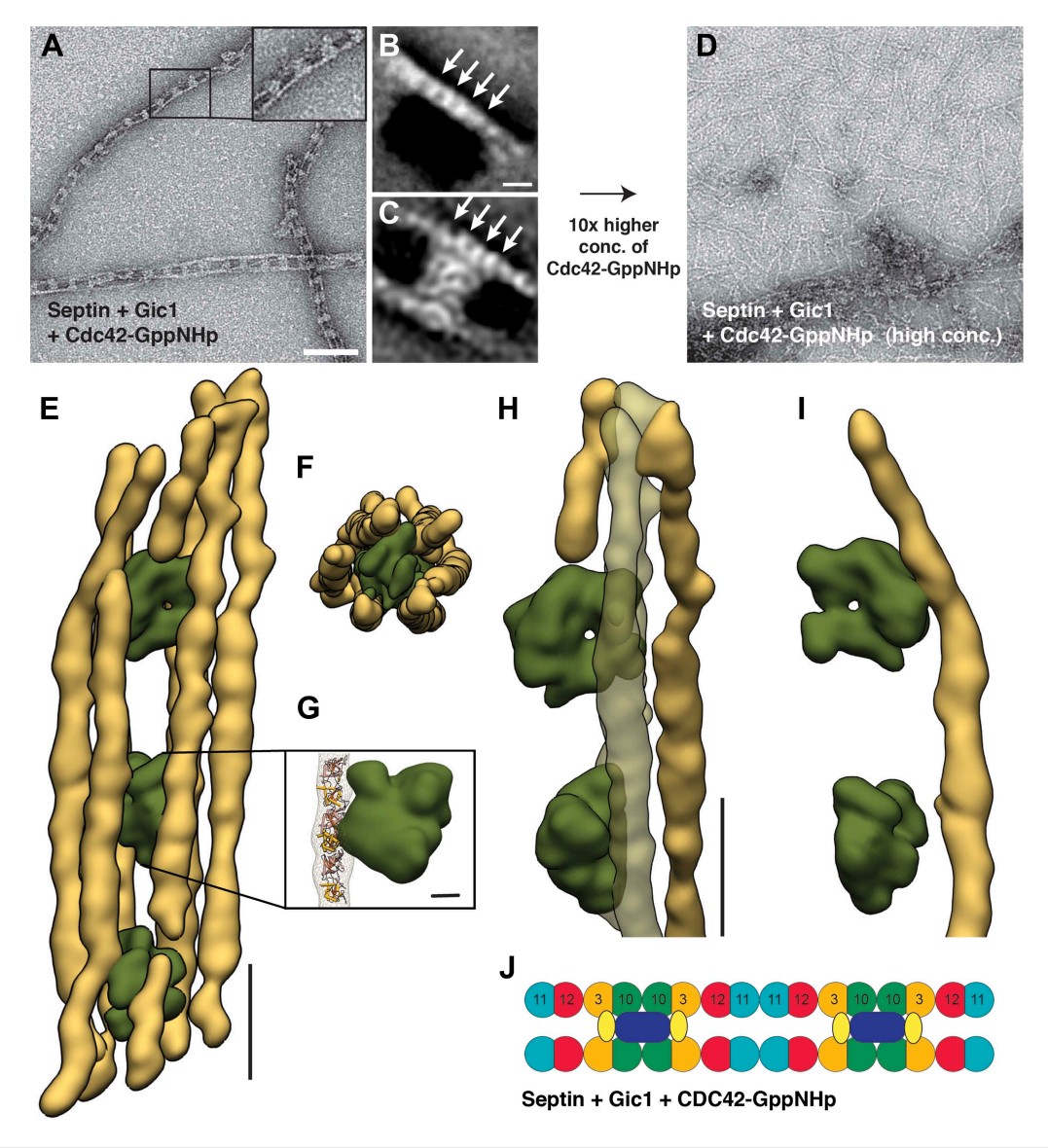

**Figure 6**. Cdc42-GppNHp binds specifically to Gic1 and dissociates septin-Gic1 complexes. (**A**) Representative EM image of negatively stained septin-Gic1-Cdc42-GppNHp complexes. The concentration of Gic1 and Cdc42-GppNHp used for filament preparation is 0.5 µM. Scale bar, 100 nm. (**B** and **C**) Representative class averages of negatively stained septin-Gic1-Cdc42-GppNHp complexes with focus on septin filaments and the cross-bridges, respectively. Arrows indicate single septin proteins. Scale bar, 10 nm. (**D**) Same as (**A**) but at 10× higher concentration of Cdc42-GppNHp. (**E–G**) Side (**E**) and top view (**F**) of a septin-Gic1-Cdc42-GppNHp complex. The septin filaments and Gic1-Cdc42-GppNHp cross-bridges are depicted in gold and green, respectively. Scale bar, 20 nm. (**G**) The crystal structure of the human SEPT2/6/7 complex (PDB 2QAG) was manually fit into the EM structure. Scale bar, 5 nm. (**H** and **I**) Side views of the septin-Gic1-Cdc42-GppNHp 3D structures. To allow a better observation of the septin filament interaction with Gic1, part of the septin filaments have been omitted. Scale bar, 20 nm. (**J**) Model of the septin-Gic1-Cdc42-GppNHp complex. Cdc42-GppNHp is depicted as yellow ovals.

which in turn polymerize. Besides activating many other proteins involved in septin recruitment and ring formation Cdc42-GTP recruits its effector Gic1 to the bud site (*Figure 16C*). Gic1, which has a high affinity for Cdc10, binds, scaffolds and stabilizes septin filaments (*Figure 16D*). Gic1-Cdc42-GTP-septin cables generated in this manner form a ring at the bud neck. Since our data show that the process of Gic1-septin cable formation does not necessarily require Cdc42-GTP, we propose

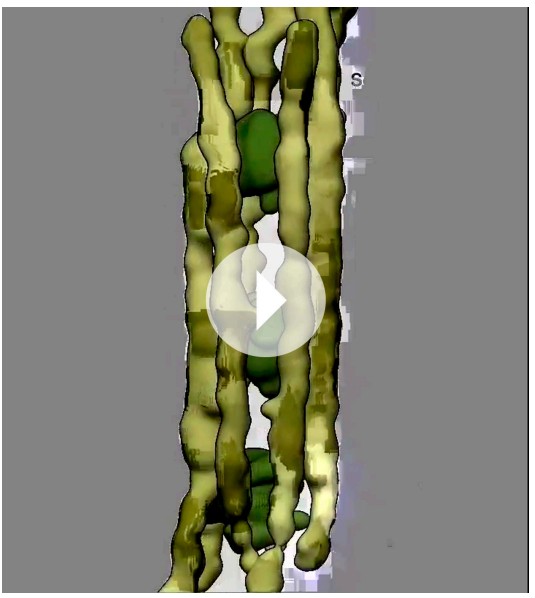

**Video 5**. Video of the 3D reconstruction of the septin-Gic1-Cdc42-GppNHp complex derived from tomograms of vitrified samples.

that upon GAP-supported GTP hydrolysis, Cdc42-GDP, which has a much lower affinity to Gic1 than Cdc42-GTP, dissociates from the Gic1-septin complex (*Figure 16E*) and recruits more septin octamers to the bud site (*Figure 16F*).

At later stages of the cell cycle the local Cdc42-GTP concentration increases at the bud neck, probably caused by an up-regulation of Cdc24 and/or identical spatial localization (*Butty et al., 2002*; *Goryachev and Pokhilko, 2008*). As a result, Cdc42-GTP binds to Gic1-septin complexes and competes with Cdc10 for Gic1, which finally results in the disassembly of the complex and thereby septin ring rearrangement (*Figure 16G*). Gic1 will then be either degraded or relocated. After GTP hydrolysis, Cdc42-GDP in the absence of Gic1 specifically binds to septin filaments and dissociates them (*Figure 16H*). Thus, septins can be reassembled and reused for the next cycle, as was observed by McMurray and Thorner when tracking labeled septins through several cell divisions (*McMurray and Thorner, 2009*). The model elegantly explains the cycles of GTP loading and hydrolysis by Cdc42 that were observed by Gladfelter et al. during budding (*Gladfelter et al., 2002*). In conclusion, Gic1 and Cdc42 in combination with many other factors, such as the nucleotide state of septins, specific interaction with

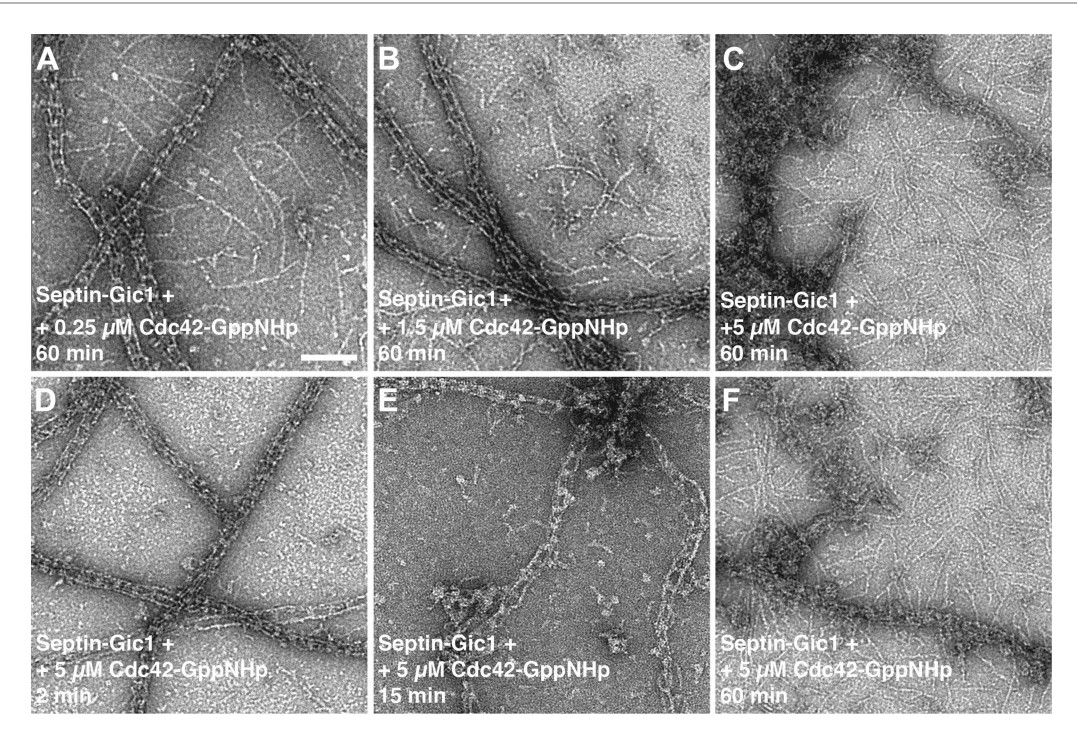

**Figure 7**. Time- and concentration-dependent interaction of Cdc42-GppNHp with the septin-Gic1 complex. (**A–C**) Representative EM images of the septin-Gic1 complex incubated for 60 min with (**A**) 0.25 μM, (**B**) 1.5 μM or (**C**) 5 μM of Cdc42-GppNHp. (**D–F**) Representative EM images of the septin-Gic1 complex incubated with 5 μM of Cdc42-GppNHp for (**D**) 2 min, (**E**) 15 min and (**F**) 60 min. Scale bar, 100 nm.

**Figure 8**. Cdc42-GDP does not interact with septin-Gic1 complexes. (**A**) Representative EM image of negatively stained septin-Gic1-Cdc42-GDP complexes. Scale bar, 100 nm. (**B**) Same as (**A**) but at 10× higher concentration of Cdc42-GDP. (**C** and **D**) Representative class averages of negatively stained septin-Gic1-Cdc42-GDP complexes with focus on septin filaments and the cross-bridges, respectively. Arrows indicate single septin proteins. Scale bar, 10 nm.

lipids and protein posttranslational modifications regulate septin recruitment, ring formation and disassembly. Most importantly, Cdc42 does not only act as a regulator, but seems to be also involved in septin recruitment.

## Materials and methods

### Plasmid construction

Construction of yeast two-hybrid plasmids of septins were previously described (*Farkasovsky et al., 2005*). In order to construct plasmids coding for the LexA and AD fusions with Gic1 fragments of different size, *gic1*(310-942) (primers *GIC1*-N3: CCAAGGATCCATGTTCAAAAAAAAGGACCTGTTG TCGAGG and *GIC1*-C1: CCAAGTCGACGGTATTTCGAGGAGTACTAGTTTC) and *gic1*(670-942) (primers *GIC1*-N4: CCAAGGATCCGATTTGGAAATGACCTTGGAAGAC and *GIC1*-C1: CCAAGTCGACGGTATTT CGAGGAGTACTAGTTTC) were amplified by using yeast chromosomal DNA as a template and the Expand High Fidelity PCR system (Roche, Mannheim, Germany). The PCR products were digested with *Bam*HI and *Sal*I and fragments were introduced between the *Bam*HI and *Sal*I sites of pEG202 or pJG4-5 vectors. The same PCR products were also used in the construction of expression plasmids pFM812 (*gic1*(310-942) in pEGST with C-terminal His$_6$-tag) or pFM562 (*gic1*(670-942) in pGEX4T-3). All plasmid constructs were confirmed by sequencing.

### Protein purification

The yeast septin complex was expressed and purified as described earlier (*Farkasovsky et al., 2005*). In order to study the interaction of Gic1 with septin filaments in vitro, we first expressed Gic1 recombinantly in *E. coli*. Since the full-length protein aggregated during expression, we tested different constructs and could obtain sufficient amounts of stable non-aggregating protein only after deleting the N-terminal 103 amino acids (*Figure 4*). Gic1(104-314) (in the text referred to as Gic1) contains the CRIB domain, which is essential for its interaction with Cdc42, and the C-terminus, which, because of its homology to the Borg BD3 domain, might be important for Gic1 binding to septins.

For the bacterial expression of Gic1(104-314), the plasmid pFM812 was transformed into the *E. coli* strain BL21 (DE3) Rosetta (Merck KGaA, Darmstadt, Germany). The cells were grown in TB medium, supplemented with ampicillin and chloramphenicol at 37°C and induced by addition of 0.2 mM IPTG at an optical density of OD$_{600}$ = 0.6. After 8 hr at 28°C, cells were harvested by centrifugation, resuspended in isolation buffer IB1 (25 mM NaHPO$_4$ pH 7.8, 5% glycerol, 0.3 M NaCl, 1 mM MgCl$_2$, 5 mM ß-mercaptoethanol, 10 mM imidazole, complete protease inhibitors [Roche], 0.2 mM PMSF) and disrupted by using a microfluidizer (Microfluidics Co., Westwood, MA, USA). After high-speed centrifugation at 100000×*g*, the supernatant was incubated with 50 ml (V$_t$) Ni-NTA-Sepharose (Qiagen, Hilden, Germany), washed with 300 ml buffer IB2 (25 mM NaHPO$_4$ pH 7.8, 5% glycerol, 0.5 M NaCl, 1 mM MgCl$_2$, 5 mM ß-mercaptoethanol, 50 mM imidazole, 5 mM ATP, complete protease inhibitors,

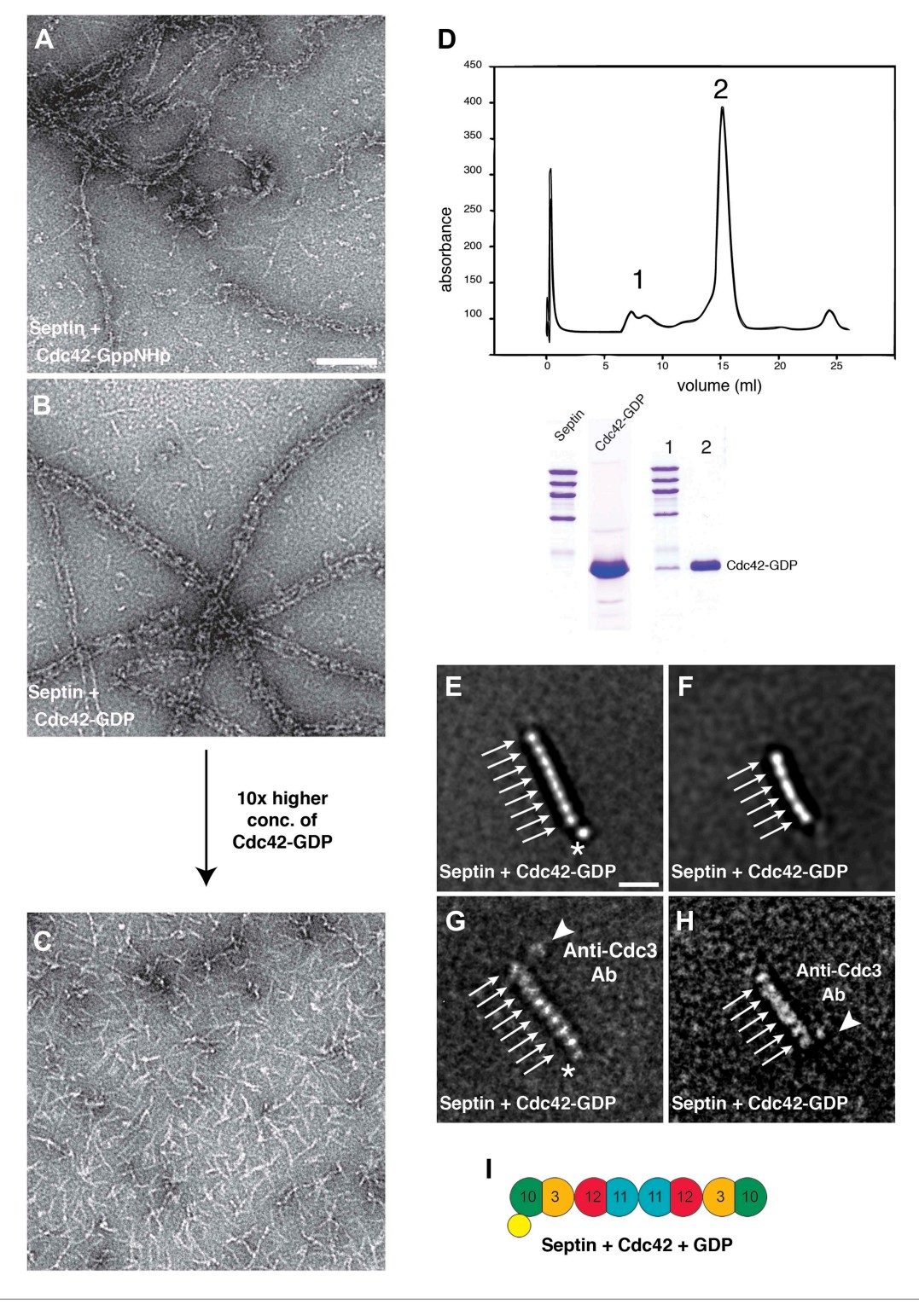

**Figure 9**. Cdc42-GDP binds specifically to Cdc10 and dissociates septin filaments. (**A**) Representative EM image of negatively stained septin complexes incubated with Cdc42-GppNHp. Scale bar, 100 nm. (**B**) Representative EM image of negatively stained septin-Cdc42-GDP-complexes. (**C**) Same as (**B**) but at 10× higher concentration of Cdc42-GDP. (**D**) Gel filtration chromatography and SDS-PAGE of the septin-Cdc42-GDP complex. (**E** and **F**) Representative class averages of septin-Cdc42-GDP complexes. The asterisk indicates the additional density corresponding to Cdc42-GDP. Arrows indicate single septin proteins. Scale bar, 10 nm. (**G** and **H**) Representative class averages of

*Figure 9. Continued on next page*

*Figure 9. Continued*

the septin-Cdc42-GDP complex labeled with antibody against Cdc3. The asterisk indicates the additional density corresponding to Cdc42-GDP. Arrows indicate single septin proteins. The triangle indicates the antibody. (**I**) Model of the septin-Cdc42-GDP complex. Cdc42-GDP is depicted as yellow circle.

0.2 mM PMSF) and with 100 ml of buffer IB3 (25 mM NaHPO$_4$ pH 7.8, 5% glycerol, 0.3 M NaCl, 1 mM MgCl$_2$, 5 mM ß-mercaptoethanol). Gic1 was eluted with 300 mM imidazole in buffer IB3 and the GST-tag was cleaved using thrombin at 4°C. Then, the GST and the undigested fusion protein were removed by glutathione sepharose column chromatography (GE Healthcare, Buckinghamshire, UK). Gic1 was concentrated and further purified on a Superdex S200 column (GE Healthcare, Buckinghamshire, UK). Expression and purification of Cdc42(G12V) was performed as previously described (**Rudolph et al., 1998**). Due its intrinsic GTPase activity, Cdc42 is usually GDP-bound after purification. In order to exchange GDP to GppNHp or to remove residual GTP, 5 mM EDTA (5 times the MgCl$_2$ concentration) and 20 times excess of the desired nucleotide over the protein were added to Cdc42 and incubated at room temperature for 2 hr. Subsequently, the protein was concentrated using Amicon Ultra-4 Centrifugal Filters with a cut-off of 10 kDa and washed with the gel filtration buffer devoid of EDTA and nucleotide (150 mM NaCl, 20 mM Tris-HCl pH 7.5, 1 mM MgCl$_2$).

## Filament preparation and antibody labeling

For septin filament production, septin oligomers in a high-salt buffer (500 mM NaCl, 1 mM MgCl$_2$, 50 mM Tris-HCl pH 7.5, 1 mM DTT) at a final concentration of 0.3 µM were dialyzed overnight at 4°C against a low-salt buffer (100 mM NaCl, 20 mM Tris-HCl pH 7.5, 1 mM DTT). In the case of Gic1-septin complexes, septin oligomers (final concentration of 0.3 µM) were mixed with Gic1 (final concentration of 1.5 µM) in a high-salt buffer and dialyzed as described above. For antibody

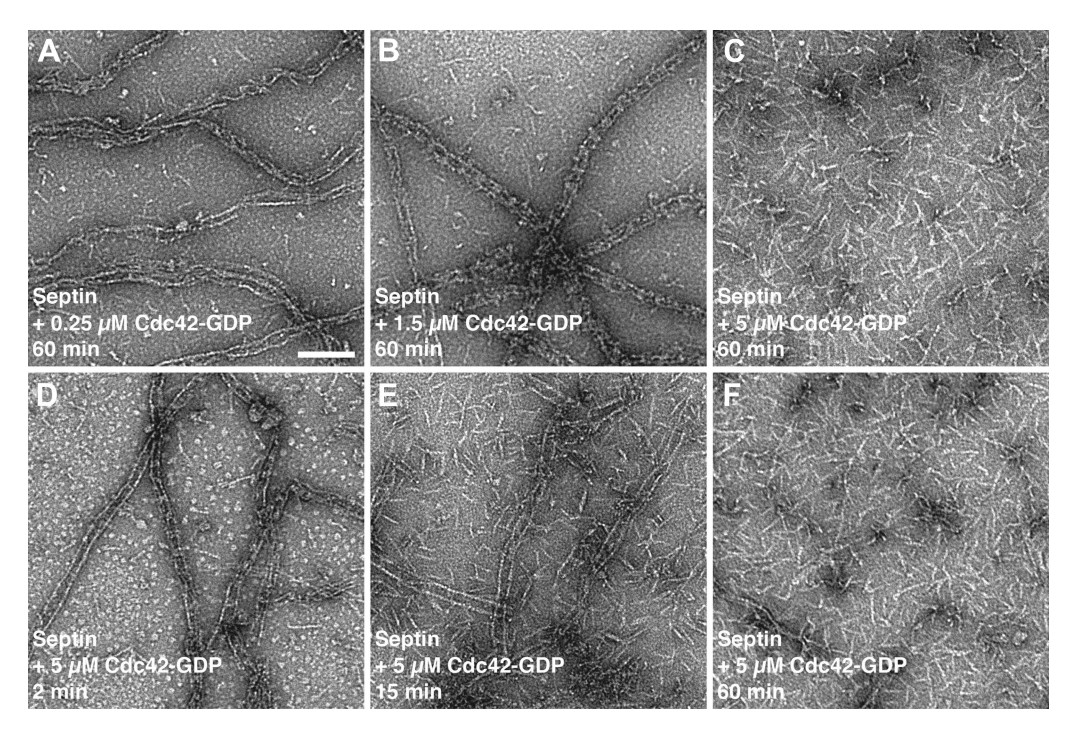

**Figure 10**. Time- and concentration-dependent interaction of Cdc42-GDP with septin filaments. (**A–C**) Representative EM images of septin filaments incubated for 60 min with (**A**) 0.25 µM, (**B**) 1.5 µM or (**C**) 5 µM of Cdc42-GDP. (**D–E**) Representative EM images of septin filaments incubated with 5 µM of Cdc42-GDP for (**D**) 2 min, (**E**) 15 min and (**F**) 60 min. Scale bar, 100 nm.

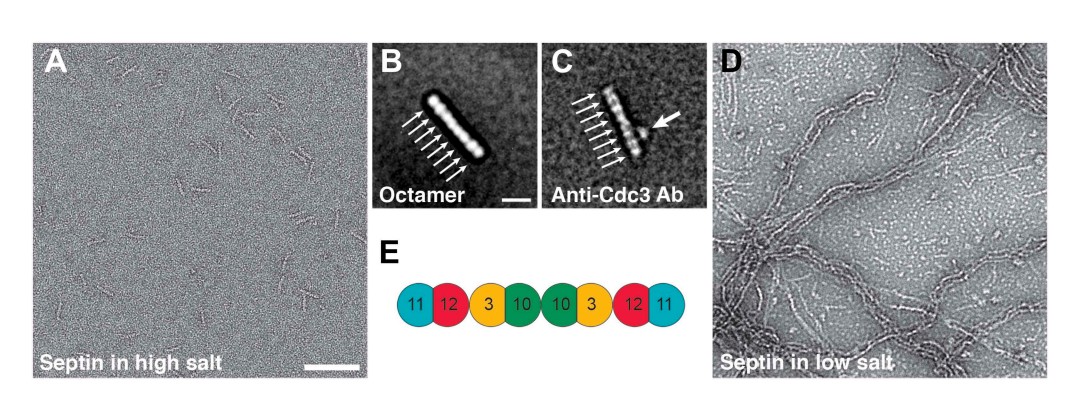

**Figure 11**. Septin polymerization depends on the ionic strength. (**A** and **B**) Representative EM images and class averages of septin complexes at high-salt (500 mM NaCl) labeled with antibody against Cdc3 (**C**) and low-salt (100 mM NaCl) (**D**) conditions. Arrows indicate single septin proteins. Scale bars, B = 100 nm, D = 10 nm. (**E**) Model of the septin complex based on the known sequential order of septin filaments. The G- and the N/C-interfaces are indicated by straight and circular interfaces between circles, respectively.

decoration, 5 µl of polyclonal antibodies against Cdc11 (Santa Cruz Biotech, Heidelberg, Germany) and Cdc3 (gift from Dr Michael Knop, ZMBH, Heidelberg) (1:100) were added to 20 µl of a sample containing the filaments or the septin octamers and incubated overnight at 4°C. For studies involving Cdc42, 0.1 µM of the septin octamer and 0.5 µM of Gic1 were used. Cdc42-GppNHp and Cdc42-GDP were used at the same concentration as Gic1 or at higher concentrations (as indicated in the figures) and incubated for different time intervals (*Figures 7 and 10*).

## Gel filtration chromatography

For gel filtration analyses of binding between non-polymerizing septin, Gic1 and Cdc42, 1 mg of each protein of the desired complex was mixed and incubated for 15 min at 4°C. Then, 500 µl of the solution was injected into Superdex S200 (GE Healthcare, Buckinghamshire, UK) column. The sample was run at 0.4 ml/min with a buffer containing 100 mM NaCl, 20 mM Tris-HCl pH 7.5, 1 mM DTT and 1 mM $MgCl_2$.

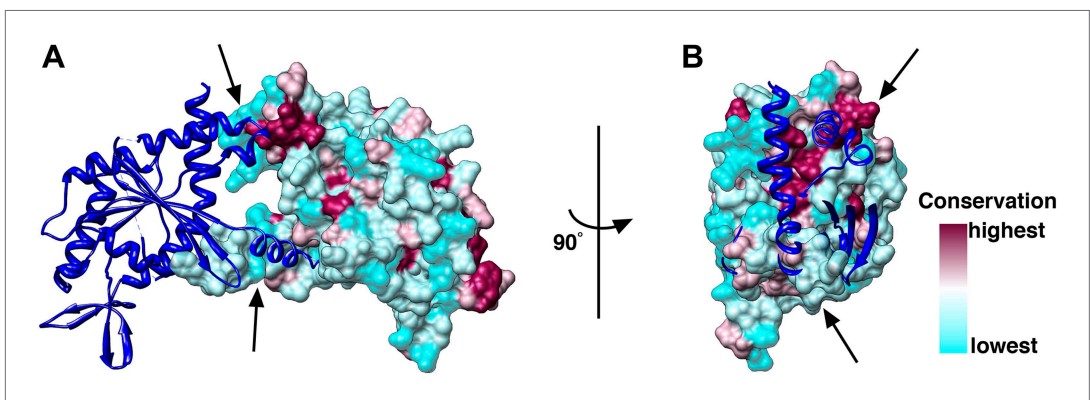

**Figure 12**. Conservation of the N/C interface between Cdc3, Cdc10, Cdc11 and Cdc12. (**A** and **B**) Side (**A**) and end-on views (**B**) of the N/C interface between two Cdc10 septins. Homology models of Cdc3, Cdc10, Cdc11 and Cdc12 using the SEPT2 structure (PDB 2QA5) as reference were calculated using Phyre2 (*Kelley and Sternberg, 2009*). The models were aligned in Chimera (*Pettersen et al., 2004*) using the SEPT2/6/7 complex (PDB 2QAG) as reference. The sequences of Cdc3, Cdc10, Cdc11 and Cdc12 were aligned using ClustalW (*Larkin et al., 2007*) and the conservation between the four septins was mapped on their surfaces.

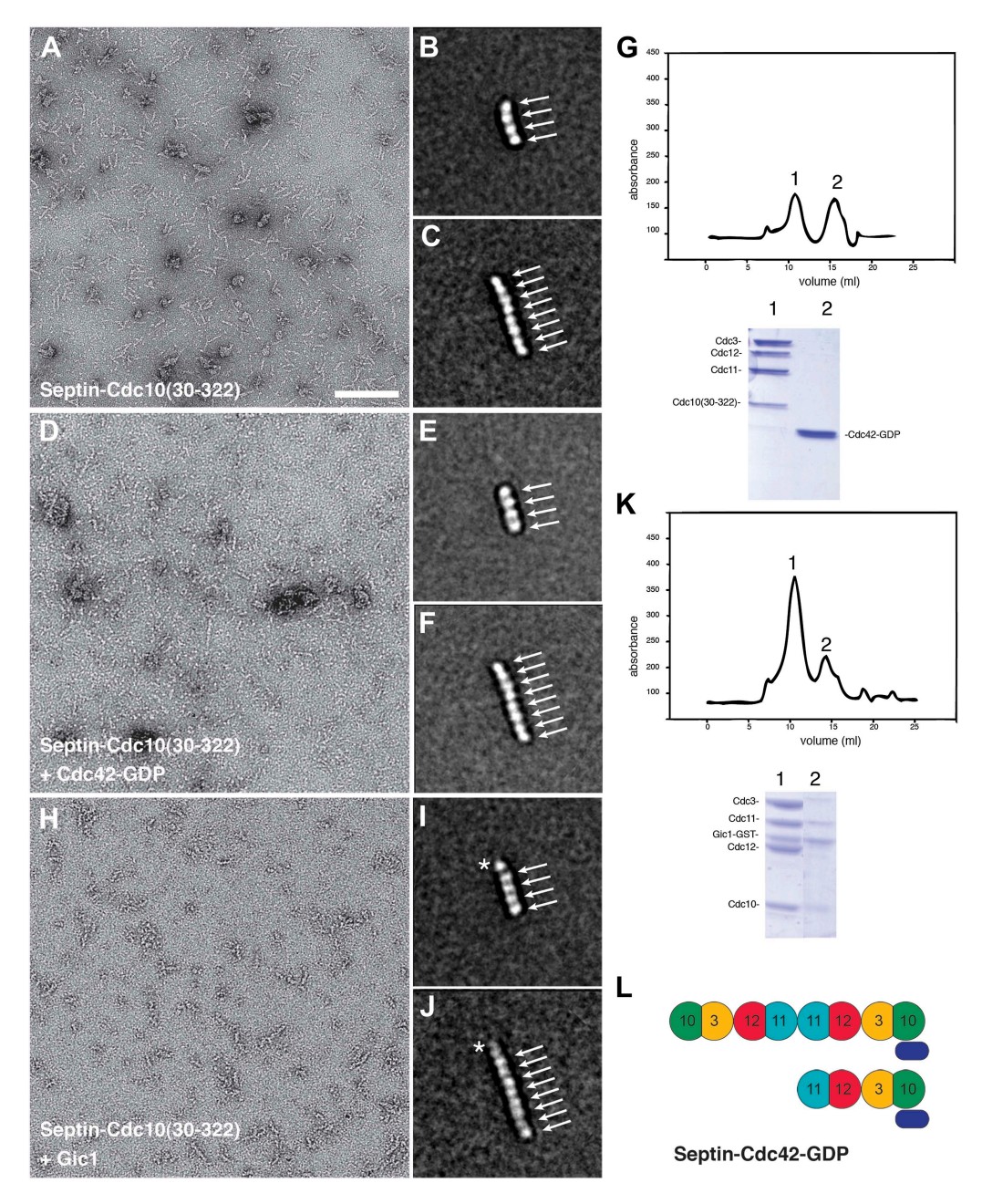

**Figure 13**. Gic1 but not Cdc42-GDP binds to polymerization impaired septin complexes. (**A–C**) Representative EM image and class averages of polymerization-impaired septin complexes containing Cdc10(30-322). Scale bars, 100 nm in (**A**) and 10 nm in (**B**). (**D–F**) Representative EM image and class averages of polymerization-impaired septin complexes containing Cdc10(30-322) + Cdc42-GDP. (**G**) Gel filtration chromatography and SDS-PAGE of septin-Cdc10(30-322) and Cdc42-GDP. (**H–J**) Representative EM image and class averages of septin-Cdc10(30-322)-Gic1 complexes. The asterisk indicates the additional density corresponding to Gic1. (**K**) Gel filtration chromatography and SDS-PAGE of the septin-Cdc10(30-322)-Gic1 complex. (**L**) Model of the septin-Cdc42-GDP complex. Gic1 is depicted as blue oval. Arrows indicate single septin proteins.

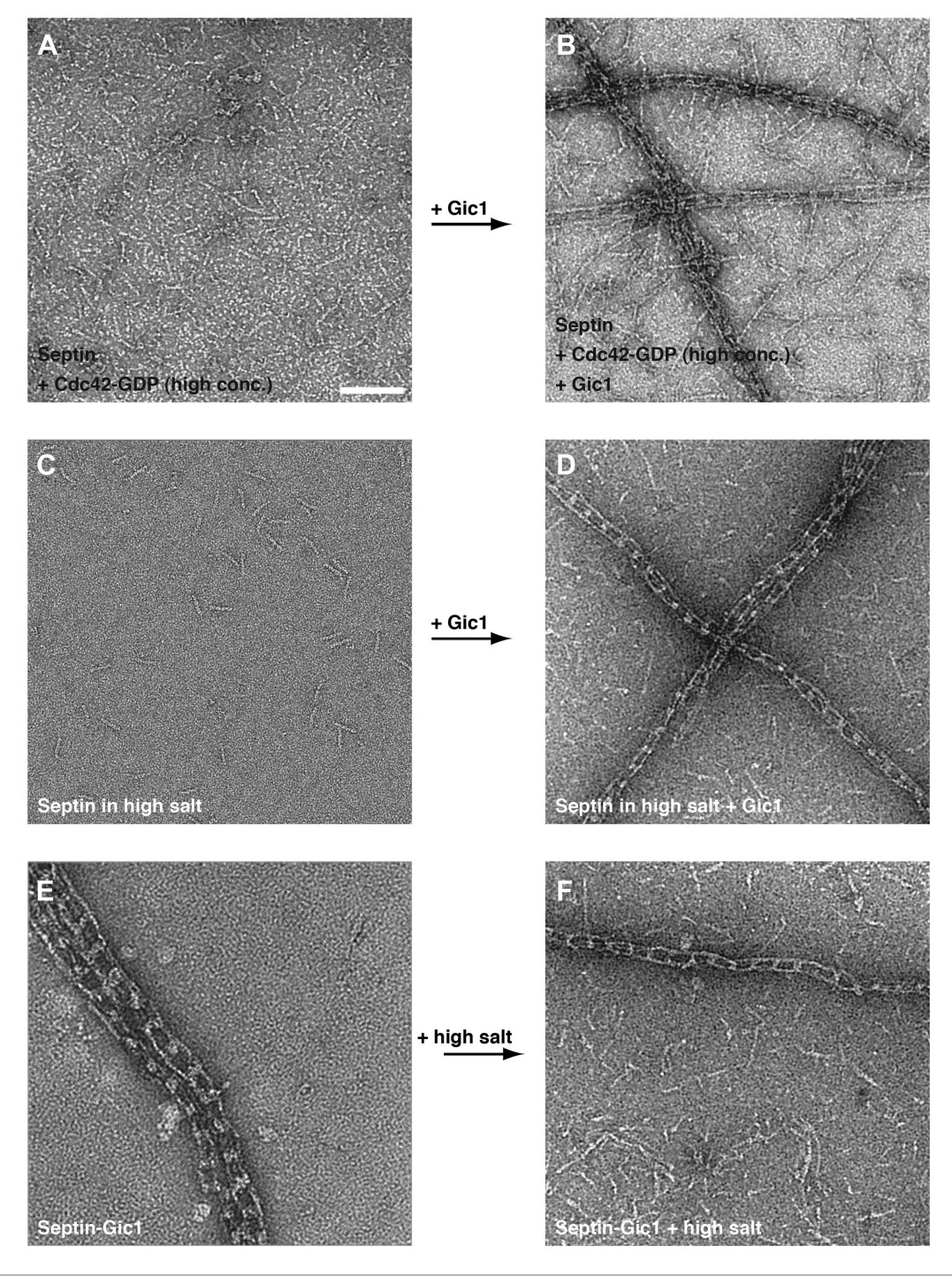

**Figure 14**. Gic1 stabilizes septin filaments. (**A–D**) Representative EM images of septin-Cdc42-GDP complexes and septin octamers in high-salt buffer before (**A** and **C**) and after incubation with Gic1 (**B** and **D**), respectively. Scale bar, 100 nm. (**E** and **F**) Representative EM images of the septin-Gic1 complex before (**E**) and after increasing the salt concentration (**F**), respectively.

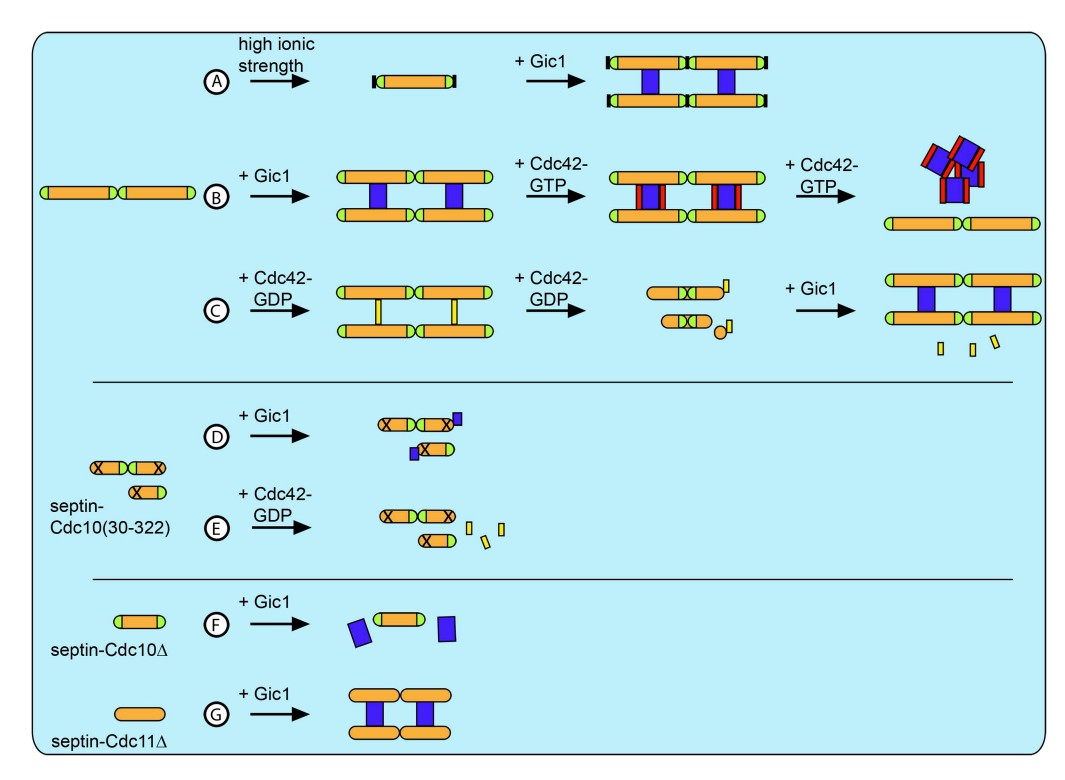

**Figure 15**. Schematic overview of all results. (**A**) At high ionic strength septin filaments disassemble into octamers (**Figure 11**). However, when Gic1 is added, septin-Gic1 filament cables are formed even at high-salt concentrations (**Figure 14**). (**B**) Gic1 binds specifically to Cdc10 and thereby scaffolds and stabilizes septin filaments forming long filament cables (**Figures 1 and 2**). Cdc42-GTP binds specifically to Gic1 resulting in a Cdc42-GTP-Gic1 complex (**Figures 5 and 6**). However, at higher Cdc42-GTP concentrations, the Gic1-septin interaction is negatively influenced and results in the dissociation of the complex (**Figure 6**). (**C**) Cdc42-GDP interacts with Cdc10 and binds specifically to septin filaments (**Figure 9**). However, at higher Cdc42-GDP concentrations, the complex dissociates into octamers with Cdc42-GDP bound to Cdc10 and Cdc10-less hexamers (**Figure 9**). Gic1 displaces Cdc42-GDP and septin-Gic1 filament cables are formed (**Figure 14**). (**D** and **E**) The polymerization of septin complexes containing Cdc10(30-322) is impaired. (**D**) Gic1 still binds to Cdc10, however, does not cross-bridge complexes and septins do not polymerize (**Figure 13**). (**E**) Cdc42-GDP does not bind to septin-Cdc10(30-322) (**Figure 13**). (**F**) Gic1 does not bind to polymerization-impaired septin-Cdc10Δ complexes (**Figure 2**). (**G**) Gic1 binds to polymerization-impaired septin-Cdc11Δ complexes resulting in septin polymerization and formation of septin-Gic1 filament cables (**Figure 2**). Septins are depicted as orange rods. Green caps indicate Cdc11. Gic1 is depicted as blue rectangle. Cdc42-GTP and Cdc42-GDP are depicted as red and yellow rectangles, respectively. The N-terminal truncation of Cdc10 is marked by a cross and the destabilization of the Cdc11-Cdc11 N/C interface is indicated by a black block.

## Yeast two-hybrid assay

Two-hybrid studies were performed using the LexA-based system as described previously (*Sirajuddin et al., 2009*). The yeast strain EGY48 was co-transformed with pEG202-based and pJG4-5-based plasmids. The reporter plasmid pSH18-34 was used for the quantitative ß-galactosidase assay. Three independent isolates of each strain were tested on minimal medium in absence of leucine or presence of X-gal, respectively.

## Adsorption to lipid monolayer

To obtain more details on the structure of septin-Gic1 complexes, we formed single railroad tracks by untangling the bundles on a lipid monolayer, which we then studied by EM and SPA. Since Cdc3 carries a His$_6$-tag, the filaments can be adsorbed to a lipid monolayer containing Ni-NTA-lipids. To obtain single-stranded filaments, 30 μl of the samples were adsorbed to a lipid

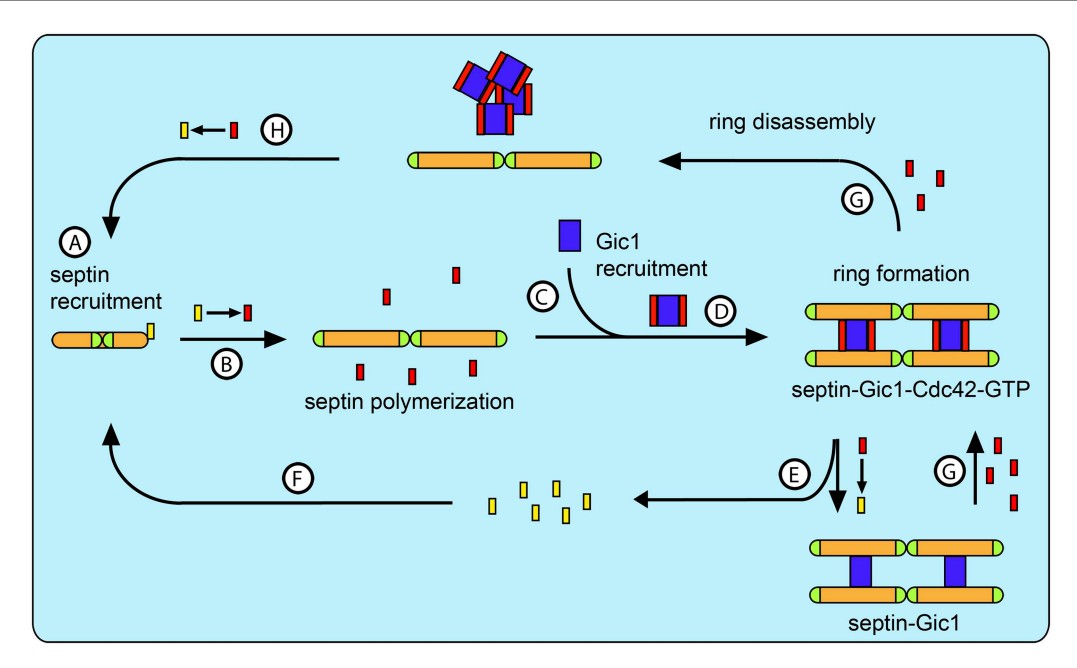

**Figure 16**. Model for septin recruitment, ring formation and disassembly. (**A**) Cdc42-GDP recruits septin complexes to the bud site. (**B**) At the bud site Cdc24 catalyzes the nucleotide exchange of Cdc42, which recruits its effector Gic1. (**C**) Septins polymerize. (**D**) Gic1 scaffolds and stabilizes septin filaments and forms septin-Gic1-Cdc42-GTP filament cables that are used for building the septin ring. (**E**) Cdc42-GTP is not necessary for the stability of the filament cables and upon GTP hydrolysis dissociates from the septin-Gic1 complexes. (**F**) Cdc42-GDP recruits more septin complexes to the bud site. (**G**) During cell division the local concentration of Cdc42-GTP increases by the up-regulation of Cdc24. This leads to a dissociation of the septin-Gic1-Cdc42-GTP filament cables and the septin ring disassembles. (**H**) After Cdc42-GAP catalyzed hydrolysis, Cdc42-GDP binds to the septin filaments and disassembles them to octamers. Septins are depicted as orange rods. Green caps indicate Cdc11. Gic1 is depicted as a blue rectangle. Cdc42-GTP and Cdc42-GDP are depicted as red and yellow rectangles, respectively.

monolayer composed of 0.5 mg/ml of DOGS-NTA:DOPC at a molar ratio of 1:3 and incubated for 1 hr at 4°C. The monolayer was transferred to a carbon-coated grid and then negatively stained as described below.

To prove that the $His_6$-tags of the proteins are not responsible for the entangling of the filaments, we performed additional experiments. Both Gics and septins have been reported to interact strongly with PIP2 (*Orlando et al., 2008*; *Bertin et al., 2010*). We therefore immobilized septin-Gic1 filaments on lipid monolayers containing PIP2 instead of Ni-NTA lipids. The filaments adsorbed to the grid and bundles were 'untangled' comparably to that seen in the experiments with Ni-NTA lipids (*Figure 17*), indicating that the interaction of the septin-Gic1 filaments, respectively, is not $His_6$-tag-induced.

## Negative stain electron microscopy and image processing

Conventional negative staining was performed as previously described (*Bröcker et al., 2012*). In brief, samples were applied onto freshly glow-discharged, carbon-coated copper grids. The sample was left for 1 min on the grid before blotting and staining with uranyl formate (0.7% wt/vol).

All images of negatively stained samples were taken with a JEOL JEM 1400 electron microscope equipped with a $LaB_6$ filament at an acceleration voltage of 120 kV. Electron micrographs were taken in minimal dose mode at a magnification of 50,000× and a defocus of 1–2 μm. Negatives (Kodak S0-163 film) were scanned with a Heidelberg Tango drum scanner with 2419 dpi resolution yielding a pixel size of 4.5 Å on the specimen level. Alternatively, images were recorded with a 4k × 4k CMOS camera F-416 (TVIPS) at a calibrated magnification of 67,200×, resulting in a pixel size of

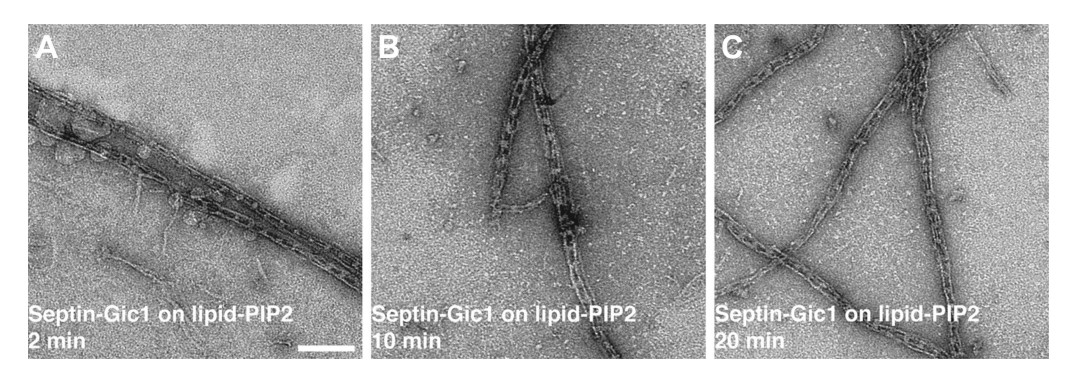

**Figure 17**. Septin-Gic1 complexes immobilized on a PIP2-containing lipid monolayer. (**A–C**) Representative EM image of negatively stained septin-Gic1 complexes immobilized on a PIP2-containing monolayer. Scale bar, 100 nm.

2.32 Å/pixel. Single particles were manually selected using boxer (*Ludtke et al., 1999*). To analyze septins and Gic1 in their non-filamentous state, 4461 particles of septin octamers, 199 particles of septin octamers labeled with anti-Cdc11 antibody and 1282 and 2195 particles of non-polymerizing septin-Cdc10(30-322) tetramers and octamers were selected. In the same way, 14,407 particles of septin-Cdc42-GDP, 451 particles of septin-Cdc42-GDP labeled with anti-Cdc3 antibody, 1152 particles of septin-Cdc42-GDP + GTP were selected. 3,169 particles of septin + GTP, 188 particles

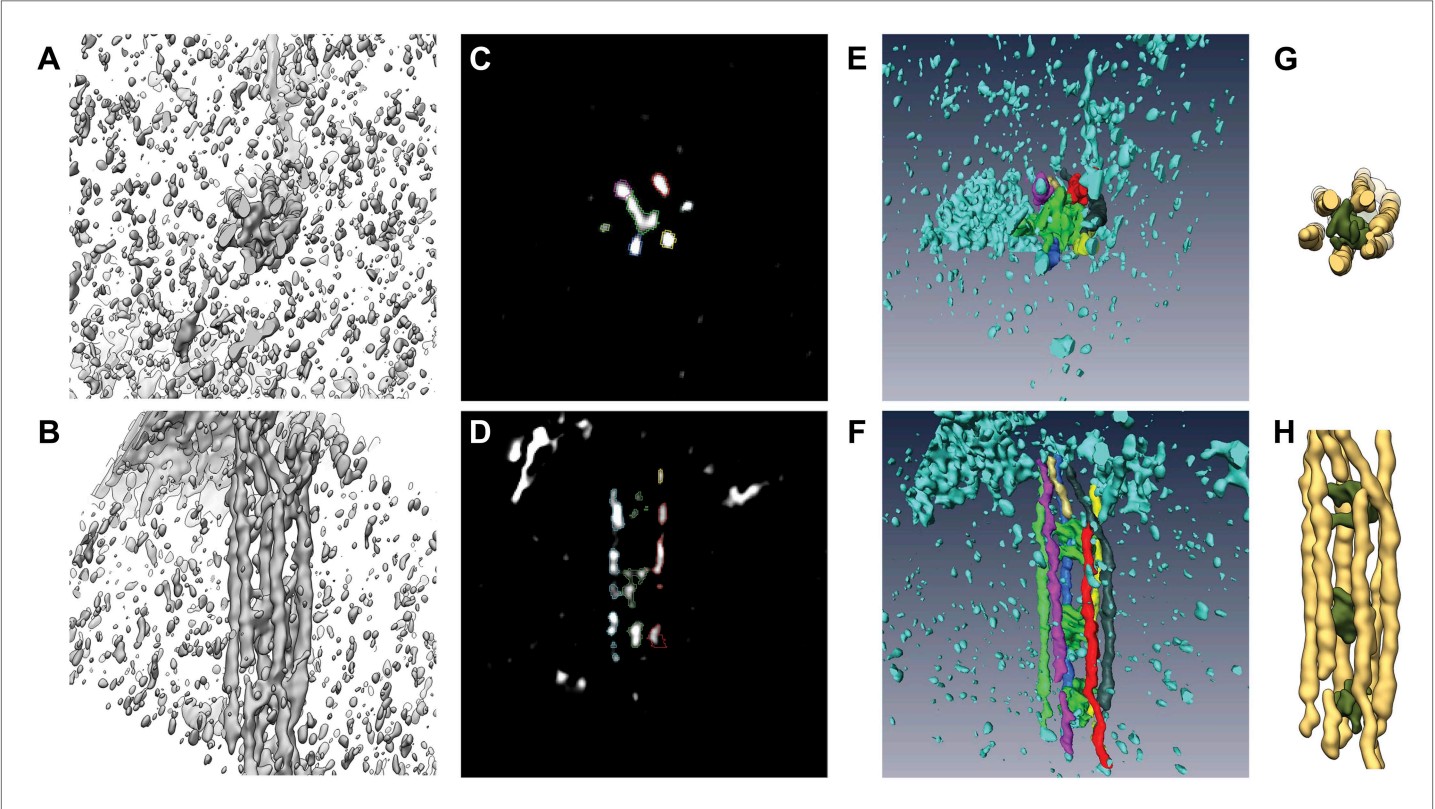

**Figure 18**. Processing of subtomograms. (**A** and **B**) Top and side view of a representative raw subtomogram filtered to 30 Å, respectively. (**C** and **D**) Segmentation in Amira (*Stalling et al., 2005*). Top and side view of representative slices with selected densities, respectively. (**E** and **F**) Top and side view of three-dimensionally rendered segments in Amira, respectively. (**G** and **H**) Top and side view of masked raw densities filtered to 40 Å using the Amira derived segments as masks.

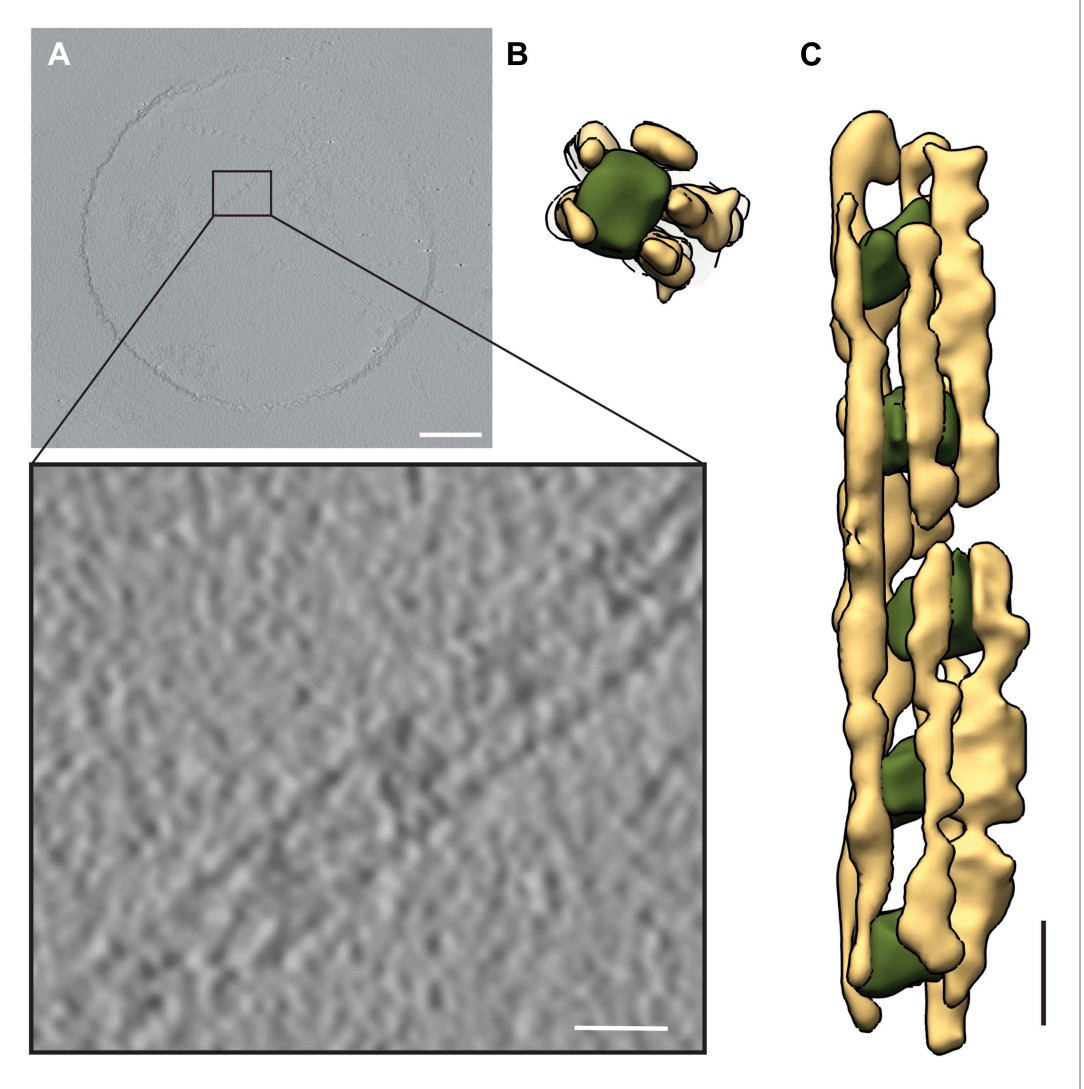

**Figure 19**. Tomography of septin-Gic1-Cdc42 complexes running parallel to the tilt-axis. (**A**) Central slice of representative tomogram of septin-Gic1-Cdc42 complexes. A railroad-like complex running parallel to the tilt axis was extracted (see inset). Scale bars, 200 nm. (**B** and **C**) Top (**B**) and side view (**C**) of a septin-Gic1-Cdc42-GppNHp complex. The septin filaments and Gic1-Cdc42-GppNHp cross-bridges are depicted in gold and green, respectively. Scale bar, 20 nm.

of septin + GTP labeled with anti-Cdc3 antibody, 281 particles of septin + GTP and anti-Cdc11 antibody were selected. To analyze septin and Gic1 in filamentous structures, two different sets of particles were selected. The first set focused on the septin filaments (sf) between two Gic1 cross-bridges and the second set in the middle of the Gic1 cross-bridge (gc). 2,228 sf particles and 1471 gc particles of the septin-Gic1 complex, 1971 sf particles and 1561 gc particles of the Cdc11Δ mutant filaments, 3979 sf particles and 3837 gc particles of the septin-Gic1-Cdc42-GppNHp complex and 180 sf particles of the septin-Gic1 complex labeled with anti-Cdc11 antibody were selected, respectively.

Single particles were aligned and classified using reference-free alignment and k-means classification procedures implemented in SPARX (*Hohn et al., 2007*). Briefly, images were normalized to the same mean and standard deviation and band-pass filtered. Images were then centered, subjected to 2D reference-free rotational alignment (sxali2d) and k-means classification (sxk_means), with approximately 100-150 images per class. The images were then further aligned and classified by several

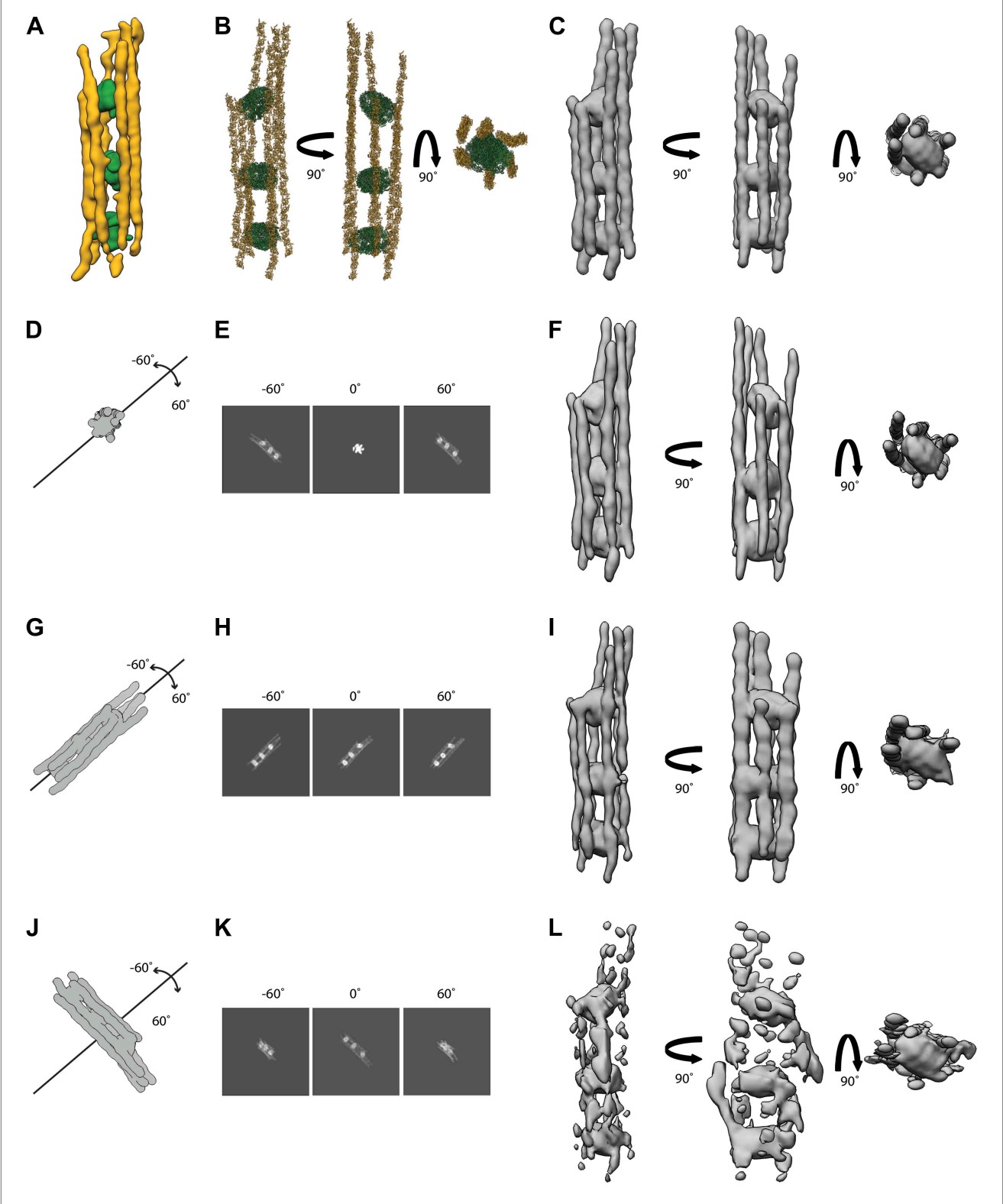

**Figure 20**. Simulations of electron tomograms of septin-Gic1-Cdc42-GppNHp complexes. (**A**) Side view of a septin-Gic1-Cdc42-GppNHp subtomogram. The septin filaments and Gic1-Cdc42-GppNHp cross-bridges are depicted in gold and green, respectively. (**B**) Model of a septin-Gic1-Cdc42-GppNHp complex obtained by fitting the crystal structure of the mammalian septin trimer (PDBid: 2QAG, gold) and GROEL/GROES (PDBid: 1AON, green) into (**A**), shown in three different orientations. (**C**) Simulated EM density map of (**B**) at a resolution of 45 Å. (**G**–**L**) Simulations of electron tomograms obtained

*Figure 20. Continued on next page*

*Figure 20. Continued*

by tilting the model shown in (**C**) in the range of ±60 in 2° increments with its long axis running parallel to the beam (**D**), parallel (**G**) and perpendicular (**J**) to our microscope's tilt axis, respectively. (**E**, **H**, and **K**) Corresponding projections at −60°, 0°, 60° and (**F**, **I**, and **L**) the resulting simulated tomograms, respectively. Note that the tomograms shown in (**I** and **L**) are obviously affected by missing wedge artifacts, whereas the tomogram in (**F**) (long axis of the molecule parallel to the beam axis during tilting) is only slightly stretched in comparison to the original model (**C**).

rounds of multireference alignment (sxmref_ali2d), where only high quality classes were used as references, followed by k-means classification. Classification was performed within an elongated mask including the respective septin density, expanded by 5 pixels. For analysis of antibody binding, all members of class averages showing additional density were merged and subjected to further rounds of alignment and classification, with approximately 15-20 images per class. Shown are characteristic class averages with the lowest intra-class variance in the region of antibody binding.

## Cryo electron tomography (cryo-ET) and image processing

For cryo-ET of the septin-Gic1 and septin-Gic1-Cdc42-GppNHp complexes, 2 µM of septins and 10 µM of Gic1 with or without 10 µM Cdc42-GppNHp were used, respectively. The septin-Gic1 and septin-Gic1-Cdc42-GppNHp complexes were mixed with 5 nm colloidal gold particles. 4 µl aliquots of each preparation were applied to a glow discharged C-flat holey carbon grid (Protochips Inc.) and plunge-frozen in liquid ethane using a Cryoplunge3 (Gatan Inc.). Images were collected with a JEOL JEM 3200FSC TEM equipped with an 8k × 8k pixel TVIPS CMOS camera (F-816) at an acceleration voltage of 200 kV and a magnification of 85,470×. An in-column omega energy filter was used to improve image contrast by zero-loss filtering with a slit-width of 15 eV.

Tilt series were collected at a defocus of ~ 4-5 µm, covering the range of ±60 in 2° increments and a dosage of about 1e−/Å$^2$ per image. Images were then reduced by 4 × 4 pixel averaging resulting in a pixel size of 7.3 Å. Data were processed using the IMOD software package (*Kremer et al., 1996*). Gold particles were tracked as fiducial markers to align the stack of tilted images, and tomograms were reconstructed by weighted back-projection.

Selected sub-tomograms were segmented using Amira (*Stalling et al., 2005*) and rendering was performed in Chimera (*Pettersen et al., 2004*). The segmentation of tomographic reconstructions was performed by manually tracing structural features through sequential slices of the tomograms. Regions of high density between filaments were assigned to Gic1 or Gic1-Cdc42 respectively.

After visual inspection of the first tomograms and subsequent careful segmentation, it became immediately clear that septin-Gic1 complexes are highly heterogeneous concerning their overall arrangement, diameters and even number of septin-filaments, which excluded the possibility of subtomogram averaging.

Furthermore, the majority of septin cables reside in a preferred side-view orientation perpendicular to the beam direction (*Figure 3*; *Video 1*). Although such tomograms gave us clear hints about the overall arrangement of septin cables and revealed clear differences between septin-Gic1 and septin-Gic1-Cdc42-GppNHp, the missing wedge artifacts caused the broadening of the septin filaments, making their exact tracing difficult in all slice-directions. However, these initial attempts revealed bending as a characteristic feature of the septin-Gic1 cables (*Figure 3*).

On grids with relatively thick ice we observed filaments, which changed their orientation and ran parallel to the beam direction at 0° tilt. In some cases, short cables were completely embedded in ice in a top view orientation (*Video 2 and 3*). Since the missing wedge artifact in this geometry causes only the elongation of the septin filaments, their exact tracing was straightforward, but the cables were often too short. We therefore scanned thousands of positions in several grids for both samples (septin-Gic1 and septin-Gic1-Cdc42-GppNHp) to find complexes of sufficient length and recorded two tomograms of such regions of septin-Gic1 and three tomograms of septin-Gic-Cdc42-GppNHp complexes. Each tomogram contained 7–10 septin-Gic1 or septin-Gic1-Cdc42-GppNHp cables running parallel to the beam at 0° tilt. We extracted all separate cables as subtomograms and processed them as described above. The quality of these raw subtomograms was extremely good and details that would otherwise require the technique of subtomogram averaging (such as number and shape of filaments, or interaction between filaments) were clearly discernable even without further processing (*Figure 18A–B*).

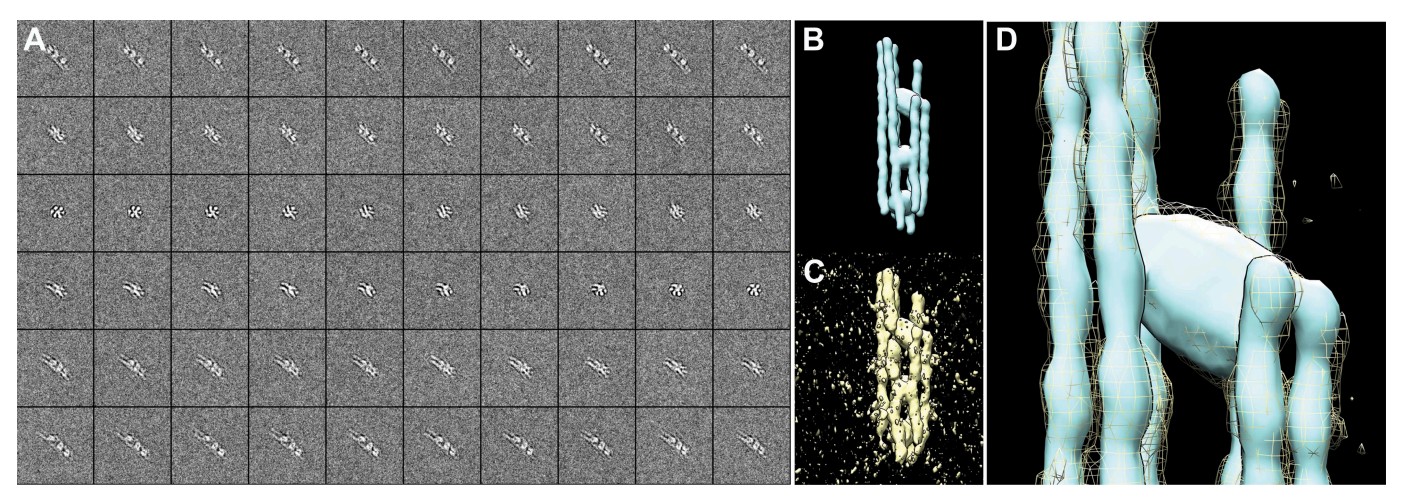

**Figure 21**. Electron tomogram simulation of septin-Gic1-Cdc42-GppNHp with CTF and noise added. (**A**) Projections of the model shown in (**B**) in the range of ±60° in 2° increments, after applying the CTF at a defocus of 4 µm and addition of noise. Note that at 0°, the long axis of the model was running in the z direction (***Figure 19D***). (**B**) Original model used for generating the reprojections (***Figure 19F***). (**C**) Reconstruction obtained by back-projecting the images shown in (**A**). (**D**) Fitting of the simulated tomographic reconstruction (yellow mesh) into the original model (cyan surface).

We segmented the subtomograms using Amira (***Figure 18C–F***). The resulting segments were then binarized, expanded, gauss filtered and used as masks to extract the respective density from the raw subtomograms. The extracted densities were low-pass filtered to 40 Å (***Figure 18G–H***). Therefore, the images shown in ***Figures 3, 6 and 18G–H*** do not represent the typical renderings of manual segmentations, but the extracted raw densities using masks obtained by manual segmentations.

To prove that the observed organization of septin-Gic1 or septin-Gic1-Cdc42 complexes is independent of the geometry during image recording, we also processed septin-Gic1-Cdc42 complexes running perpendicular to the beam and parallel to the tilt axis (***Figure 19A–C***). As expected and described above the septin filaments are broadened and flat. However, the structure clearly shows that the septin-Gic1-Cdc42 complexes have the same organization as the complexes depicted in ***Figure 3E–G*** and ***Figure 6E–G***, therefore ruling out that our observations are influenced by the geometry of the complexes.

To determine the effect of the missing wedge on our structure we performed tomography-simulations using an idealized railroad-like structure model. The idealized model of a septin-Gic1-Cdc42-GppNHp complex (***Figure 20B–C***) was obtained by fitting several copies of the crystal structure of the mammalian septin trimer (PDBid: 2QAG) into the density of septin filaments and placing of GROEL/GROES (PDBid: 1AON) into the density corresponding to Gic1-Cdc42-GppNHp cross-bridges in one of our reconstructions (***Figure 20A***).

We then simulated electron tomograms by tilting the model shown in ***Figure 20C*** in the range of ±60 in 2° increments with its long axis running parallel to the beam, and parallel and perpendicular to our microscope's tilt axis, respectively (***Figure 20D,G,J***).

All tomograms are affected by missing wedge artifacts. However, the complexes running parallel to the tilt axis are better resolved (***Figure 20E–F,H–I***) than the ones running perpendicular to the tilt axis (***Figure 20K–L***). Because of the longitudinal appearance of the septin-Gic1-Cdc42 complexes, tomograms with complexes running parallel to the beam result in a reconstruction that is more similar to the original model (***Figure 20E–F***) compared to complexes running perpendicular to the beam (***Figure 20H–I***). The filaments are fully comparable to the filaments of the original model. Only the densities of GROEL/GROES (simulating Gic1-Cdc42 cross-bridges) (***Figure 20F***, shown in green) show an increase in diameter along the z direction by ~14%.

Moreover, to simulate a more 'close to reality' situation, the same procedure was repeated, this time after adding noise and applying CTF to the projections (***Figure 21A–B***). The volume was then filtered to 45 Å (***Figure 21C***) and compared again to the simulated model (***Figure 21D***). Even without further processing of this simulated tomogram, both volumes show a high degree of similarity in all

aspects with a cross-correlation coefficient of 0.95, further suggesting that tomograms taken under similar conditions are fully sufficient to describe the basic architecture of the filaments.

## Filament preparation and antibody labeling

For septin filament production, 0.3 µM of septin hetero-oligomers (wild-type, EGFP-tagged, Cdc10Δ, Cdc11Δ or Cdc10(30-322) mutants) in a high-salt buffer (500 mM NaCl, 1 mM MgCl$_2$, 50 mM Tris-HCl pH 7.5, 1 mM DTT) were dialyzed overnight at 4°C against a low-salt buffer (100 mM NaCl, 20 mM Tris-HCl pH 7.5, 1 mM DTT). In order to form the septin-Gic1 complexes, 1.5 µM of Gic1 was included during dialysis. Since Gic1 forms dimers and septin octamers contain two Cdc10 subunits, the molar ratio (Cdc10:Gic1) is 1.25:1. For antibody decoration, 5 µl of polyclonal antibodies against Cdc11 (Santa Cruz Biotechnology Inc.) and Cdc3 (a gift from Michael Knop, DKFZ-ZMBH, Heidelberg, Germany) (diluted 1:100) were added to 20 µl of a sample containing either septin-Gic1 complex or the septin octamers (generated by GTP, Cdc42-GDP or both) and incubated overnight at 4°C to allow efficient binding.

In order to evaluate the effect of Cdc42 on septin complexes, 0.1 µM of septin octamers and 0.5 µM of Gic1 were used. Cdc42-GppNHp and Cdc42-GDP were used at a concentration equal to Gic1 or at 10 times higher concentrations and incubated for different time intervals (*Figures 7 and 10*). To assess the effect of nucleotides on septin complexes, 2.4 mM of GMP, GDP, GTP or GppNHp were added to 0.5 µM of septin filaments dialyzed alone or with 2.5 µM of Gic1 and incubated for 16 hr at 4°C. The same sample was dialyzed to remove the residual GTP and the generated octamers were incubated with 25 µM of Cdc42-GDP and incubated at 4°C for 16 hr. For cryo-ET of the septin-Gic1 and septin-Gic1-Cdc42 complexes, 2 µM of YSC and 10 µM of Gic1 with or without 10 µM Cdc42-GppNHp were used, respectively.

## Data deposition

The coordinates for the EM structures have been deposited in the EM Data Bank under accession codes EMDB-2504 and EMDB-2505.

## Acknowledgements

We are grateful for A Wittinghofer for initiating the project and for useful comments on the manuscript. We thank I Vetter for stimulating discussions.

## Additional information

### Funding

| Funder | Grant reference number | Author |
|---|---|---|
| Deutsche Forschungsgemeinschaft | RA 1781/1-1 | Stefan Raunser |
| Max Planck Society | | Yashar Sadian, Christos Gatsogiannis, Oliver Hofnagel, Roger S Goody, Stefan Raunser |
| VEGA (Vedecká Grantová Agentúra, Ministerstva školstva Slovenskej republiky) | 2/0050/11 | Csilla Patasi, Marian Farkašovský |

The funders had no role in study design, data collection and interpretation, or the decision to submit the work for publication.

### Author contributions

YS, Acquisition of data, Analysis and interpretation of data, Drafting or revising the article; CG, CP, OH, Acquisition of data, Analysis and interpretation of data; RSG, Analysis and interpretation of data, Drafting or revising the article; MF, Conception and design, Acquisition of data, Analysis and interpretation of data; SR, Conception and design, Analysis and interpretation of data, Drafting or revising the article

# Additional files

## Major datasets

The following datasets were generated:

| Author(s) | Year | Dataset title | Dataset ID and/or URL | Database, license, and accessibility information |
|---|---|---|---|---|
| Sadian Y, Gatsogiannis C, Patasi C, Hofnagel O, Goody RS, Farkašovský M, Raunser S | 2013 | Data from: The Role of Cdc42 and Gic1 in the Regulation of Septin Filament Formation and Dissociation | http://www.ebi.ac.uk/pdbe/entry/EMD-2504 | Publicly available at Electron Microscopy Data Bank (EMDB). |
| Sadian Y, Gatsogiannis C, Patasi C, Hofnagel O, Goody RS, Farkašovský M, Raunser S | 2013 | Data from: The Role of Cdc42 and Gic1 in the Regulation of Septin Filament Formation and Dissociation | http://www.ebi.ac.uk/pdbe/entry/EMD-2505 | Publicly available at Electron Microscopy Data Bank (EMDB). |

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
