## [Decision Letter]

Thank you for sending your work entitled “The Role of Cdc42 and Gic1 in the Regulation of Septin Filament Formation and Dissociation” for consideration at *eLife*. Your article has been favorably evaluated (subject to revisions outlined below) by a Senior editor and 3 reviewers, one of whom is a member of our Board of Reviewing Editors.

The Reviewing editor and the other reviewers discussed their comments before we reached this decision, and the Reviewing editor has assembled the following comments to help you prepare a revised submission.

In this in vitro study with recombinant proteins, the role of the guanine nucleotide binding protein Cdc42 and the regulator Gic1-septin filament formation and dissociation is elucidated. Using a combination of negative stain electron microscopy and cryo electron tomography, along with manipulations of the septin complex or the nucleotide state of Cdc42, a dual role of Cdc42 was discovered. In the GTP-state, Cdc42 leads to disassembly of the Gic1-septin complex, whereas in the GDP-state, Cdc42 leads to further disassembly of the septin complex in absence of Gic1. Moreover, the cryo EM reconstructions show that Gic1 can bind to multiple septin filaments in a heterogeneous fashion, accounting for flexibility of septin filaments during cell division and cytokinesis.

Previous studies have linked Gic1/2 and Cdc42 nucleotide cycling to septin assembly in vivo but really how Cdc42 and its myriad of effectors actually lead to septin ring and collar formation remain mysterious. While Gic1/2 function has been genetically linked to septin organization and shown to physically associate with septin complexes, precisely what the Cdc42 effector does to regulate septins is unknown. Similarly, for Cdc42 itself, which is essential for septin assembly, it has not been determined if Cdc42 can bind septins and directly promotes filament assembly, filament bundling and/or if it simply indirectly recruits septins to the bud site.

The electron micrographs are striking and represent some of the most detailed glimpses at purified septins presented to date. However, there are significant concerns about the quality and resolution of the EM tomograms that need to be addressed, along with the other concerns outlined below, before a decision can be made.

Overall, the data indicate that 1) Gic1 can bundle septins through specific interactions at Cdc10, 2) Cdc42-GDP can bind septin octamers (through a surprisingly differently-ordered octamer than what has been shown for pure septin complex), 3) Cdc42-GTP changes how Gic1 associates with septin filaments and may itself associate with Gic1 bound to filaments, 4) high concentrations of Cdc42-GTP can lead to Gic1 dissociation for filaments and therefore ultimately destabilize the filaments. These are intriguing findings and do represent the first clean reconstitution of regulated septin assembly. Thus, the studies in this work should stimulate future studies to confirm that this is what is happening in vivo.

Required revisions:

1) The authors employ cryo-ET to view these fascinating assemblies 3D. However, lack of details precludes confidence in the displayed models. The authors need to detail the number of data sets collected and analyzed, post-processing and segmentation procedures, and should provide slices in X Y directions. Moreover, from reviewing the movies kindly provided the claim of isotropic resolution is inadequate. The resolution in Z direction is severely diminished, and this is the long axis of the bundles used. There is no way of telling what type of artifacts this geometry will introduce. If the authors want to prove that their conclusions are supported by the data, the authors will need to come up with some more convincing arguments. Maybe the authors can apply some simulations with idealized models or make use of the bundles with their long axis running in the other directions (x, y). As it stands there is some probability that the data is over interpreted, at least to some extent. Furthermore, because of the absence of averaging, the authors are looking at single motifs in a fairly noisy background, which is not helping in building confidence for their conclusions.

2) The bundles of septin-Gic complexes were “untangled” by adsorbing them to a lipid monolayer containing NTA-lipids that bind to His6-tags attached to the Cdc3 septin molecule. Do the His-tags themselves cause some aggregation in this system, i.e., could that be the origin of the entangling of the filaments? In other words, were EM studies attempted with a different method to tether them to the surface?

3) Please provide the control polymerization done in absence of fluorophore to establish the lack of interference (artifacts) induced by the addition of the EGFP-tag.

4) What is the estimated resolution of the cryo-EM tomograms in xy and z, respectively (see also point 1)?

5) What is the concentration of Cdc42-GppNHp in Figure 5?

6) The Gic components in the septin-Gic complexes appear rather heterogeneous. Do they consist of multiple Gic molecules or is Gic thought to be a flexible multi-domain protein?

7) Figure 1: The fluorescence in B looks just aggregated rather than bundled: these images raise some concern about what the relevance is of filament cables to normal septin assemblies. There is no scale bar annotation for panels C-F.

8) Figure 2: There are no scale bar annotations. There is only one spot along the fiber that is Cdc11 labeled. Please mention how many particles were used for the averaging in Figure 2. This is an important potential issue because previous work by Iwase et al. found Gic specifically complexing with Cdc12 not Cdc10.

9) The interpretation of Figure 5 that the bridges get bulkier with Cdc42 added is that Cdc42 is part of enlarged bridging structure with Gic, but this could also be Gic1 oligomerization – without further experiments this can't be excluded.

10) Figure 8: How does one know that band from gel filtration is Cdc42-GDP and not a contaminant from bacteria? Could a Western blot also be included?

11) It is perplexing that filaments breaking at Cdc10 interface in presence of excess Cdc42. It would be interesting in seeing this result validated in some other manner.

12) For easy reference please keep the color coding of Cdc3-EGFP, Cdc10, Cdc11 and Cdc12 consistent with the one published in [1].

13) In [1], cross-bridges between pairs of filaments were observed emanating from Cdc12 and Cdc3, suggested to be their CTEs. Here, Sadian et al. only observe cross bridges if Gic or Cdc42 are added (through Cd10). The authors need to discuss the underlying source of the conflicting results.

[Editors’ note: before acceptance, the following revisions were also requested.]

Thank you for resubmitting your work entitled “The Role of Cdc42 and Gic1 in the Regulation of Septin Filament Formation and Dissociation” for further consideration at *eLife*. Your revised article has been evaluated by a Senior editor, a member of the Board of Reviewing Editors (Axel Brunger), and two reviewers. We thank the authors for addressing many of the concerns raised by the reviewers and the Reviewing editor. The manuscript has been improved but there are some remaining issues that need to be addressed before acceptance, as outlined below.

There are remaining significant concerns about the quality and reliability of the cryo-EM tomography results (see the comments by Reviewer 3 below). The reviewers and editors had considerable discussions about these concerns. They recognize that main conclusion that is drawn from the tomograms is that Gic1 cross-bridges more than two filaments, a result that could not be drawn from the negative stain EM images alone. The potential artifacts that are caused by the missing wedge in the EM data could have affected this particular conclusion, especially the connections between the filaments, as the authors also showed in their simulations.

Ideally, the authors should revisit their existing EM tomograms at the orthogonal orientation to show that these connections are independent of the geometry (see Reviewer 3 comments). The authors (using the geometry in Figure 19 for example) showed that there are clear extra crosslinks visible in tomograms. However, we would be prepared to accept the paper in essentially its current form if the authors would clearly point out the limitations of their current tomography study (including the potential effects of the missing wedge, lack of averaging, the estimated very low resolution of the images). The authors need to make a convincing case that the main conclusion drawn from the tomograms (more than two filaments binding to Gic1) is probably not affected by these concerns. Moreover, the authors must clearly state in the main text, the figure captions, and the conclusions that the images themselves may have gross errors.

*Comments from Reviewer 3*:

The authors have now given a much clearer picture of the tomography data and processing. Unfortunately, much of what was laid out only confirmed the reviewer's initial concerns. It is not the quality of the reconstruction that improves by changing the relationship between beam direction and tilt axis. The data in the missing wedge is missing, still. The artifacts are just along a different direction. The illusion of “improvement” is actually dangerous because it gives a false sense of quality for the reconstruction.

The authors show themselves with their simulations that the data collection strategy they use generates severe artifacts to thin structures that are perpendicular to the beam. If there are any thin connections between the filaments, it is very likely that they will not appear in the reconstruction with the chosen geometry. The modeling of the cross-links as featureless barrels is not helpful to convince the reviewer. The question is what would happen to finer features that may be attached to these barrels or emanate from/connect between the filaments.

In trying to get a quantitative assessment of the assembly features (filaments and linkers) for this geometrically challenging system, the authors should have generated tomo-series in both orthogonal orientations to provide dual axis tomograms. Such data scheme would allow assessing the extent of the missing wedge and isotropic distortions for the system and thus provide measurable values to the confidence to the described conclusions.

Furthermore it is somewhat concerning that the whole analysis is based on five reconstructions, 2 of septin-Gic1 and 3 of septin-Gic-Cdc42-GppNHp complexes. These numbers are on the lower side to draw robust conclusions, specifically as thousand of such assemblies were viewed.

In summary, the reliability of the features in the reconstructions is questionable and the presentation of the data is misleading by implying that artifacts caused by the missing wedge would be somehow overcome by using a specific geometry.

---

## [Author Response]

*1) The authors employ cryo-ET to view these fascinating assemblies 3D. However, lack of details precludes confidence in the displayed models. The authors need to detail the number of data sets collected and analyzed, post-processing and segmentation procedures, and should provide slices in XY directions*.

We admit that the description of our approach was not completely adequate, and apologize for this. We now provide more technical details in Materials and methods.

We also provide slices and the corresponding segmentations in XY and Z direction and also included a direct comparison between raw and processed subtomograms in an additional figure (Figure 18), showing that the main features of the septin-Gic1 and septin-Gic1-Cdc42 complexes described in the present study are well defined even in the noisy unprocessed subtomograms.

*Moreover, from reviewing the movies kindly provided the claim of isotropic resolution is inadequate. The resolution in Z direction is severely diminished, and this is the long axis of the bundles used. There is no way of telling what type of artifacts this geometry will introduce. If the authors want to prove that their conclusions are supported by the data, the authors will need to come up with some more convincing arguments. Maybe the authors can apply some simulations with idealized models or make use of the bundles with their long axis running in the other directions (x, y). As it stands there is some probability that the data is over interpreted, at least to some extent. Furthermore, because of the absence of averaging, the authors are looking at single motifs in a fairly noisy background, which is not helping in building confidence for their conclusions*.

To obtain an optimal coverage of the complexes during a single-tilt tomographic image acquisition, the long axis of the complexes had to run either parallel to the tilt axis or parallel to the beam axis of the microscope. In the first case, the complexes were strongly affected by the missing wedge. The filaments appeared stretched and merged partially with each other and therefore their tracing was difficult. On the other hand, in tomograms with complexes running parallel to the beam, the septin filaments were clearly defined, all filaments around Gic1 had the same diameter and their tracing was straightforward. Furthermore, the densities corresponding to Gic1 and Gic1/Cdc42, respectively were fully comparable respective their overall shape and size to tomograms with complexes running in perpendicular directions, at least at the resolution level of 40-50 Å. Thus, tomograms of complexes running in the z direction provided the best possible 3D reconstructions of these complexes. We added two additional Videos (Videos 2 and 3) showing cryo electron tomograms with septin-Gic1 complexes running parallel to the beam.

However, we are aware that these tomograms are still affected by the missing wedge along the z direction, as pointed out by the reviewers. In order to show to what extent this results in structural artifacts, we performed tomography-simulations using an idealized railroad-like structure model, as suggested. These simulations are now included in Figures 19 and 20 and described in detail in the revised manuscript.

In the case of the model running with its long axis parallel to the beam at 0˚, the filaments do not show prominent missing wedge artifacts at a resolution of 45 Å and are fully comparable to the filaments of the original model. Only the density of GROEL/GROES (simulating Gic1-Cdc42) (Figure 19, shown in green) shows an increase in diameter along the z direction by ∼14 %.

We are confident that this mild anisotropy does not lead to wrong conclusions, especially since we do not describe any specific molecular or mechanistic details and restrict our analysis of 3D-tomograms exclusively to basic low-resolution features of the complexes. Taken together, this rules out that we over-interpret our data.

*2) The bundles of septin-Gic complexes were “untangled” by adsorbing them to a lipid monolayer containing NTA-lipids that bind to His6-tags attached to the Cdc3 septin molecule. Do the His-tags themselves cause some aggregation in this system, i.e., could that be the origin of the entangling of the filaments? In other words, were EM studies attempted with a different method to tether them to the surface? Note to authors: the response to our query along with the PIP2 control should be included in the revised paper; no additional experiments are needed*.

In a classical EM grid preparation, samples are incubated on carbon coated copper grids to which they absorb non-specifically. In the case of filaments that tend to bundle, the whole grid is often covered with protein making the analysis of single filaments difficult and tedious. Septin filaments only interact with lipid monolayers if the monolayers contain special lipids. His-tagged septin filaments can be immobilized on lipid monolayers with Ni-NTA lipids and by varying the concentration of Ni-NTA lipids the concentration of septin filaments on the grid can be conveniently adjusted. This facilitates the sample preparation and is an established method in electron microscopy (Kelly et al. 2010).

However, in order to prove that the His-tags of the proteins are not responsible for the entangling of the filaments, we performed additional experiments. Both Gics and septins have been reported to interact strongly with PIP2 (29; 2). We therefore immobilized septin-Gic1 filaments on lipid monolayers containing PIP2 instead of Ni-NTA lipids. The filaments adsorbed to the grid and bundles were “untangled” comparably to that seen in the experiments with Ni-NTA lipids (Figure 17), indicating that the interaction of the septin-Gic1 filaments, respectively, is not His-tag-induced.

Another good argument against the hypothesis of a His-tag-induced interaction is that we also observed the “entangling” of septin filaments when we studied unmodified wildtype septins purified from yeast (9). The bundling of His-tagged septins is not stronger than that of the wildtype filaments. We can therefore exclude that the His-tag forces the filaments to “entangle”.

In our study, we showed that His-tagged Gic1 interacts with the septin Cdc10 (Figure 2). The His-tag on the septin filaments, however, is attached to a different septin, namely Cdc3. We can therefore also exclude a His-tag induced interaction of Gic1 and septins. In addition, we performed experiments with GST-tagged Gic1 and observed the same railroad structures as for His-tagged Gic1, indicating that the interaction between Gic1 and Cdc10 is not induced by His-tags (Figure 22).Author response image 1.Septin-Gic1-GST complexes.

Finally, the His-tags do not aggregate homogeneous Gic1 (Figure 4) or septin octamer samples (Figures 11 and 14). The entangling of septin filaments must therefore be caused by other physical and chemical interactions comparable to the bundling of actin filaments.

In summary, we can exclude artifacts induced by the His-tags.

*3) Please provide the control polymerization done in absence of fluorophore to establish the lack of interference (artifacts) induced by the addition of the EGFP-tag*.

The only experiments we have done in the presence of a fluorophore are shown in Figure 1/B. All other experiments were done without additional EGFP-tag. We point this out more clearly in the revised manuscript.

We also performed experiments with EGFP-tagged septins (see Figure 23), but did not include them in the manuscript to prevent confusion. Using EGFP-tagged proteins, we observe the same railroad track structure as in the experiments without the fluorophore. However, the density of Gic1 appears to be much larger, since EGFP localizes directly to Cdc3 and adds density to the cross-bridge between the filaments.Author response image 2.Complexes between Gic1 and septin filaments containing Cdc3-EGFP.**(A-B)** Representative EM images of negatively stained septin filaments containing Cdc3-EGFP polymerized by dialysis alone **(A)** or together with Gic1 **(B)**. Scale bar, 100 nm. **(C-D)** Representative class averages with focus on the Gic1 cross-bridges **(C)** and the septin-EGFP filaments **(D)**. Arrows indicate single septin proteins. Scale bar, 10 nm. (**E**) Model of the septin-Gic1 complex based on the known sequential order of septin filaments (1). The G- and the N/C-interfaces are indicated by straight and circular interfaces between circles, respectively. EGFP is indicated as light green ovals.

*4) What is the estimated resolution of the cryo-EM tomograms in xy and z, respectively (see also point 1)*?

Despite much progress in electron tomography, quantitative assessment of resolution in single tomograms remains a problematic issue. A resolution estimate is given for only 25 of the 60 tomograms deposited in the EMDB. For 7 of these 25 tomograms, although a value is given, it is not described how the authors determined it. For 5 tomograms, the resolution was estimated by visual inspection regarding certain known features in the tomograms. For 7 tomograms, the resolution was estimated by the first zero-crossing of the CTF in the 0˚ image (however, this criteria does not distinguish between signal and noise, and is therefore just a measure for how good the resolution could be in principle). For the remaining specimens, the resolution was determined by electron diffraction (this is only possible for ordered structures such as 2D crystals or helices). A cross validation-based criterion exists for electron tomography (Cardone et al. 2005), but because of many problems it extremely underestimates the resolution of the tomograms and is therefore not used by the tomography community.

Therefore, due to the absence of a standard method, we avoid estimating the resolution of our tomograms. However, segmentation of our tomograms was straightforward after filtering down the raw tomograms to 30 Å. The first zero-crossing in the CTF in the 0˚ images of the various tomograms is at about 35-45 Å. Moreover, a visual comparison, between the densities of septin filaments in our tomograms and down-filtered crystal structures of human septin (see also Figures 18, 19 and 20), suggests a resolution of 40-60 Å. Thus, for our sub-tomograms shown in the present study, we expect a resolution in the xy direction of 40-60 Å (at lower resolutions, septin filaments are expected to appear as featureless straight rods). Furthermore, based on an elongation factor of 1.14 that we derived from our simulation studies, we expect a 5-10 Å decrease of resolution in the z direction.

*5) What is the concentration of Cdc42-GppNHp in*
Figure 5?

As described in the Materials and methods, Cdc42-GppNHp was added at equal concentrations as Gic1 for filament preparation, i.e., 0.5 µM and at 10x higher concentrations, i.e. 5 µM. To prevent any confusion we included the values now in the legend to Figure 6 (formerly Figure 5).

Taken into account that each yeast cell contains around 1000 copies of septin octamers and Gic1 (Ghaemmaghami et al. 2003), the estimated concentration of septins or Gic1 in a yeast cell is around 0.05 µM. However, the local concentration of Cdc42 and Gic1 in yeast (for example at the budding neck) can be much higher (11; 12). We therefore believe that our observations are physiologically relevant.

*6) The Gic components in the septin-Gic complexes appear rather heterogeneous. Do they consist of multiple Gic molecules or is Gic thought to be a flexible multi-domain protein*?

Gic1, which based on its sequence has a molecular weight of 23 kDa (Gic1(104-314)), elutes at about 49 kDa from a gel filtration column corresponding roughly to a dimer (Figure 4). In a typical Gic1 cross-bridge up to 12 Cdc10 subunits are involved (two per filament) (Figure 6). If we assume that each of them binds independently to a Gic1 dimer, we would expect that 12 Gic1 dimers assemble into a large cross-bridge of 600 kDa. In good agreement with this the average large volume of the Gic1 density in the electron tomograms indicates that it must contain multiple Gic1 molecules. The heterogeneity of the Gic1 cross-bridges indicates that the number of Gic1 molecules varies between cross-bridges. We included an additional paragraph in the revised manuscript.

*7)*
Figure 1*: The fluorescence in B looks just aggregated rather than bundled: these images raise some concern about what the relevance is of filament cables to normal septin assemblies. There is no scale bar annotation for panels C-F*.

To get a strong signal in the light microscope high concentrations of septin and septin-Gic1 filaments were used. In addition, the images were taken with an epifluorescence microscope not with a confocal microscope. That is why the overlapping septin filaments appear aggregated. However, the extended EM study clearly shows that we observe filament cables and bundles, but no aggregates. The scale bar annotations are included in the revised manuscript.

*8)*
Figure 2*: There are no scale bar annotations. There is only one spot along the fiber that is Cdc11 labeled. Please mention how many particles were used for the averaging in*
Figure 2*. This is an important potential issue because previous work by Iwase et al. found Gic specifically complexing with Cdc12 not Cdc10*.

The scale bar annotations are included in the revised manuscript.

As explained in the legend the arrow in Figure 2 does not indicate Cdc11, but the antibodies bound to Cdc11. Since there are two adjacent Cdc11 septins, two antibodies bind instead of one as shown in Figure 2. We accidentally wrote “antibody” instead of “antibodies”. We changed this in the revised manuscript.

As stated in the Material and methods, we used 199 particles that had clearly antibodies bound for this analysis. Each class average contained 15 single particles. We state that now in the revised manuscript.

However, the number of single particles in the class average is not an important potential issue. Because of the work by Iwase et al. that indicated that Gic would bind to Cdc12, we did three independent studies to prove that Gic1 binds to Cdc10. First, we did the antibody labeling mentioned above. Second, we performed septin-Gic1 complex formation with septin-Cdc10Δ complexes that showed that Gic1 does not bind to septin-Cdc10Δ filaments although they contain Cdc12 (Figure 2). Third, we performed yeast two-hybrid assays that clearly showed that Gic1 binds to Cdc10 (Figure 2). Notably, in contrast to Iwase et al., who got a signal after 9 days, we obtained a clear-cut signal already after 2 days. We describe our studies of the Cdc10Δ and Cdc11Δ-septins now in more detail in the revised manuscript to make this point clear.

*9) The interpretation of*
Figure 5
*that the bridges get bulkier with Cdc42 added is that Cdc42 is part of enlarged bridging structure with Gic, but this could also be Gic1 oligomerization – without further experiments this can't be excluded*.

In Figure 5 (formerly Figure 4) we show that Cdc42-GppNHp binds specifically to Gic1. The gel filtration profile in Figure 4 does not indicate any kind of oligomerization of Gic1 and the Gic1-Cdc42-GppNHp complex elutes from the gel filtration column at the expected position (Figure 5). We can therefore clearly state that Cdc42-GppNHp binds to Gic1 and does not induce Gic1 oligomerization. The binding of Cdc42-GppNHp must therefore be the reason for the broadening of the cross-bridges in the case of septin-Gic-Cdc42-GppNHp complexes.

*10)*
Figure 8*: How does one know that band from gel filtration is Cdc42-GDP and not a contaminant from bacteria? Could a Western blot also be included*?

The experiment represented in Figure 9 (formerly Figure 8) was performed with pure proteins. The His-tagged proteins (Cdc3 and Gic1) and GST-tagged Cdc42 were recombinantly expressed in *E. coli* and purified by affinity chromatography to high purity. We prepared septin filaments, added Cdc42-GDP and incubated the mixture overnight prior to gel filtration.

In order to prove the purity of our proteins, we added SDS-PAGE lanes of the purified proteins to Figure 9 in the revised manuscript.

*11) It is perplexing that filaments breaking at Cdc10 interface in presence of excess Cdc42. It would be interesting in seeing this result validated in some other manner*.

In an additional experiment, we identified the Cdc42 binding site on Cdc10 validating our previous results. The following passage, including figures, has been added to the revised manuscript:

“To identify the Cdc42 binding site on Cdc10, we calculated homology models of Cdc3, Cdc10, Cdc11 and Cdc12 using the SEPT2 structure (PDB 2QA5) as a reference and mapped regions of conservation between the four structures on their surfaces…”

*12) For easy reference please keep the color coding of Cdc3-EGFP, Cdc10, Cdc11 and Cdc12 consistent with the one published in*
[1].

We changed the color coding in all figures accordingly.

*13) In*
[1]*, cross-bridges between pairs of filaments were observed emanating from Cdc12 and Cdc3, suggested to be their CTEs. Here, Sadian et al. only observe cross bridges if Gic or Cdc42 are added (through Cd10). The authors need to discuss the underlying source of the conflicting results*.

Looking at high concentrations of septin filaments we see the same thin cross-bridges that were described by [1] (Figure 24). These cross-bridges connect parallel running filaments that lie very close to each other. Using negative stain electron microscopy, they can normally only be seen when many filaments with cross-bridges at the same positions lie exactly on top of each other. Depending on the thickness of the stain, we can sometimes see them connecting two septin filaments also at lower septin filament concentration (Author response image 3B). The Gic1-mediated cross-bridges, however, are much larger and can easily be seen in negative stain electron microscopy. The stain accumulation between the Gic1 cross-bridges makes it impossible to visualize the thin cross-bridges emanating from coiled-coil interactions of the C-terminal extensions of Cdc3 and Cdc12, although they are probably still there. In addition, the resolution of the cryo electron tomography is not sufficient to visualize the thin coiled-coil cross-links between the C-terminal extensions of Cdc12 and Cdc3.

We added a passage discussing this issue in the revised manuscript.Author response image 3.Cross-bridges between bare septin filaments.**(A)** Very high concentration of septin filaments. **(B)** Low concentration of septin filaments. Note, due to negative staining the thin cross-bridges between bare septin filaments mediated by the C-terminal extensions of Cdc3 and Cdc12 can only rarely be visualized. Scale bar, 100 nm.

[Editors’ note: before acceptance, the following revisions were also requested.]

*There are remaining significant concerns about the quality and reliability of the cryo-EM tomography results (see the comments by Reviewer 3 below). The reviewers and editors had considerable discussions about these concerns. They recognize that main conclusion that is drawn from the tomograms is that Gic1 cross-bridges more than two filaments, a result that could not be drawn from the negative stain EM images alone. The potential artifacts that are caused by the missing wedge in the EM data could have affected this particular conclusion, especially the connections between the filaments, as the authors also showed in their simulations*.

*Ideally, the authors should revisit their existing EM tomograms at the orthogonal orientation to show that these connections are independent of the geometry (see Reviewer 3 comments). The authors (using the geometry in*
Figure 19
*for example) showed that there are clear extra crosslinks visible in tomograms*.

We revisited our EM tomogram at the orthogonal orientation and searched for filaments running parallel to the tilt axis, exactly as simulated in Figure 19 (now Figure 20) (see also below) and processed the tomograms as described for the other septin-Gic1 filaments (see Material and methods). As expected the septin-Gic1-Cdc42-GppNHp complexes look similar to the structures obtained from bundles that run parallel to the beam (new Figure 19). Despite missing wedge artifacts, there are clear Gic1-Cdc42 cross-bridges (“or extra crosslinks”) that connect several, in this case five, septin filaments. This proves that observation of Gic1 or Gic1-Cdc42 connections is independent of the geometry.

*However, we would be prepared to accept the paper in essentially its current form if the authors would clearly point out the limitations of their current tomography study (including the potential effects of the missing wedge, lack of averaging, the estimated very low resolution of the images). The authors need to make a convincing case that the main conclusion drawn from the tomograms (more than two filaments binding to Gic1) is probably not affected by these concerns. Moreover, the authors must clearly state in the main text, the figure captions, and the conclusions that the images themselves may have gross errors*.

We hope that our additional reconstructions and our answers to the reviewer’s comments (see below) are convincing. Additionally we included some sentences that explain the limits of cryo-ET, such as missing wedge artifacts, in the revised manuscript.

Comments from Reviewer 3:

*The authors have now given a much clearer picture of the tomography data and processing. Unfortunately, much of what was laid out only confirmed the reviewer's initial concerns. It is not the quality of the reconstruction that improves by changing the relationship between beam direction and tilt axis. The data in the missing wedge is missing, still. The artifacts are just along a different direction. The illusion of “improvement” is actually dangerous because it gives a false sense of quality for the reconstruction*.

This reviewer is correct in stating that the missing wedge is always there (i.e., missing), independent of the beam direction and tilt axis. We did not claim that there are no missing wedge artifacts. But dependent on the orientation of the complexes during tomography these artifacts have different effects on the overall computed structures of the complexes.

If the symmetry or pseudo-symmetry axis of an object is running parallel to the tilt axis of the microscope, the overall tomographic coverage of the structure improves. This is a fact that all EM users take seriously into account during the recording of single particles tomograms, especially when averaging is not possible.

From the home page of SerialEM (the most common program to acquire tilt series for electron tomography; http://bio3d.colorado.edu/SerialEM/index.html), Strategies for Cryoelectron Tomography: “There is essentially no dual-axis capability due to the low dose requirements. So, we are left with single-axis reconstructions. If the sample allows you to, taking advantage of sample orientation to the tilt-axis can become very important. The best resolution is parallel to the tilt-axis.”

We were surprised that the reviewer criticized this standard approach in the first review. However, we agreed to provide simulation experiments. They clearly show that the complexes running parallel to the tilt axis are better resolved (Figure 20) than the ones running perpendicular to the tilt axis (Figure 20). Because of the longitudinal appearance of the septin-Gic1-Cdc42 complexes, tomograms with complexes running parallel to the beam (Figure 20) result in a reconstruction that is more similar to the original model compared to complexes running perpendicular to the beam (Figure 20).

*It is not the quality of the reconstruction that improves by changing the relationship between beam direction and tilt axis*.

There must be a misunderstanding: the relationship between beam direction and tilt axis does not change during a tomography experiment. It depends on the current microscope setup and if the user does not tilt the beam during image recording (we never tilt the beam!), this relationship will not change.

*If there are any thin connections between the filaments, it is very likely that they will not appear in the reconstruction with the chosen geometry*.

As described above we revisited our EM tomogram at the orthogonal orientation and calculated a reconstruction of filaments running parallel to the tilt axis (Figure 19). As expected, we do not see any additional thin connections between the filaments.

The thin connections described in Bertin et al (1) are expected to be composed of two interacting thin coiled-coils. They are very thin and cannot be seen in 2D class averages (this work) (34) and 3D reconstructions of negatively stained septin octamers at 27 Å resolution (Lukoyanova et al. 2008).

The resolution of our reconstructions is 40-60 Å at best (as pointed out in our previous rebuttal letter, there is not yet a method to measure precisely the resolution of single tomograms). It is therefore expected that these structures are not resolved in our tomograms. Even if the thin connections were visible, their segmentation would be highly ambiguous, independent of the chosen geometry. This issue is pointed out in the revised manuscript more clearly.

*The modeling of the cross-links as featureless barrels is not helpful to convince the reviewer*.

We chose GROEL/GROES as a model because it has similar dimensions to Gic1 cross-bridges and has been extensively studied by electron microscopy.

*The question is what would happen to finer features that may be attached to these barrels or emanate from/connect between the filaments*.

As described above, finer features will not be resolved at a resolution of 40-60 Å. We have used electron tomography to study these complicated structures, not X-ray crystallography. We therefore restrict our description and conclusion to gross features. We agree with the reviewer that this would be worth analyzing in future studies, but with other techniques, in particular X-ray crystallography.

*In trying to get a quantitative assessment of the assembly features (filaments and linkers) for this geometrically challenging system, the authors should have generated tomo-series in both orthogonal orientations to provide dual axis tomograms. Such data scheme would allow assessing the extent of the missing wedge and isotropic distortions for the system and thus provide measurable values to the confidence to the described conclusions*.

Although dual axis tomography theoretically improves the quality of tomograms, it harbors many technical problems and has therefore not been routinely used by the EM community. Most electron microscopes, including ours, are not equipped with a dual axis tomography holder. Therefore one has to remove the grid from the microscope after recording the first tomogram, turn it by 90˚ and enter it again. After finding the right position the second tomogram is recorded. Besides many obvious technical difficulties such as ice contamination when removing the grid and problems with alignment, the electron dose of such tomograms is doubled, resulting in decreased resolution. Any potential improvement of our tomograms would therefore be questionable. Instead we revisited our EM tomograms at the orthogonal orientation and searched for filaments running parallel to the tilt axis. This showed that the overall structure of the septin-Gic1 complexes is independent of its orientation in the tomogram.

From the home page of SerialEM (the most common program to acquire tilt series for electron tomography; http://bio3d.colorado.edu/SerialEM/index.html), Strategies for Cryoelectron Tomography: “There is essentially no dual-axis capability due to the low dose requirements. So, we are left with single-axis reconstructions.”

From “Cryo-electron tomography: The challenge of doing structural biology in situ” (Vladan Lučić, Alexander Rigort, and Wolfgang Baumeister, JCB:Review, August 5, 2013), Dual-axis tomography: “Tilting around two tilt axes perpendicular to each other reduces missing wedge–induced distortions and improves information content. However, dual-axis tomography has rarely been used in cryo-ET due to the need to acquire an increased number of projections and difficulties with alignment....”

*Furthermore it is somewhat concerning that the whole analysis is based on five reconstructions, 2 of septin-Gic1 and 3 of septin-Gic-Cdc42-GppNHp complexes. These numbers are on the lower side to draw robust conclusions, specifically as thousand of such assemblies were viewed*.

We could provide hundreds of reconstructions with complexes running in random directions relative to the tilt axis. However, as described above and demonstrated by simulations, only reconstructions with filaments running parallel to the tilt axis or to the beam result in analyzable reconstructions. To find filaments in this orientation is very challenging. We are convinced that the number of reconstructions provided is sufficient to support our conclusions.

*In summary, the reliability of the features in the reconstructions is questionable and the presentation of the data is misleading by implying that artifacts caused by the missing wedge would be somehow overcome by using a specific geometry*.

We clearly show that we do not overcome the missing wedge artifacts. This is not possible in a single tomogram experiment. However, we show that the orientation of the longitudinal complexes during tomography definitely has an impact on the resolution and overall quality of the reconstructions. We rephrased this passage in the revised manuscript to prevent any misunderstandings.

Due to the overall methodical and resolution limitations our paper focuses exclusively on gross features of the complex. We can clearly determine the number of filaments and we can even fit the crystal structure of human septin into our reconstructions to verify that the overall dimensions are correct. Although we cannot exclude the possibility that the density corresponding to Gic1 appears more stretched due to the missing wedge artifacts, we can state that Gic1 cross-bridges several septin filaments. These are our conclusions from the cryo-ET data.

We are convinced that we have carefully interpreted, and not over-interpreted, the data presented and have avoided any overstatement. Therefore, we are confident that the conclusions we draw from our data, namely that Gic1 scaffolds more than two septin filaments, are robust and justified.